# GRAPH-OF-AGENTS: A GRAPH-BASED FRAMEWORK FOR MULTI-AGENT LLM COLLABORATION

**Sukwon Yun**[1] **Jie Peng**[1] **Pingzhi Li**[1] **Wendong Fan**[2] **Jie Chen**[3] **James Zou**[4]
**Guohao Li**[2] **Tianlong Chen**[1]
[1]UNC Chapel Hill [2]Eigent AI [3]MIT-IBM Watson AI Lab [4]Stanford University

## ABSTRACT

With an ever-growing zoo of LLMs and benchmarks, the need to orchestrate multiple models for improved task performance has never been more pressing. While frameworks like Mixture-of-Agents (MoA) attempt to coordinate LLMs, they often fall short in terms of (1) selecting relevant agents, (2) facilitating effective intra-agent communication, and (3) integrating responses efficiently. In this work, we propose Graph-of-Agents (GoA), a new graph-based framework for modeling multi-agent LLM communication. Our approach begins with node sampling, selecting only the most relevant agents by leveraging model cards that summarize each model's domain, task specialization, and other characteristics. Next, we construct edges between the selected agents by evaluating their responses against one another to determine relevance ordering. Directed message passing is then performed from highly relevant agents to less relevant ones to enhance their responses, followed by reverse message passing to refine the original responses of the more relevant agents. Finally, the updated responses are aggregated via graph-based pooling (e.g., max or mean pooling) to produce a single, unified answer. We evaluate GoA on diverse multi-domain benchmarks (MMLU, MMLU-Pro, GPQA) and domain-specific benchmarks (MATH, HumanEval, MedMCQA), with an agent pool of 6 LLMs spanning multiple domains. Surprisingly, GoA achieves superior performance using only 3 selected agents, outperforming recent multi-agent LLM baselines that utilize all 6 agents simultaneously. By adopting a graph structure, GoA offers both scalability and effectiveness through structured message passing—positioning it as a strong candidate for navigating the challenges of the ever-growing LLM zoo. Code is available at: https://github.com/UNITES-Lab/GoA.

## 1 INTRODUCTION

Too many LLMs, too many benchmarks. As the ecosystem of Large Language Models (LLMs) (Wei et al., 2022a) rapidly diversifies, researchers are increasingly overwhelmed—not just by the sheer number of models available, but also by the complexity of evaluating and combining them effectively at test time to solve complex tasks. In this era of abundant LLMs, a central challenge emerges:

> **(Q)** *Given the diversity of available LLMs, how can we design an effective playground where agents interact synergistically—leveraging strengths, compensating for weaknesses, and improving decision-making through efficient collaboration?*

As an early attempt to address this challenge, Mixture-of-Agents (MoA) (Wang et al., 2024a; Li et al., 2025) has recently been introduced as a pioneering approach that explores how leveraging multiple LLMs can enhance overall model performance through the Mixture-of-Experts (MoE) (Shazeer et al., 2017) framework. As illustrated in Figure 1 (c), MoA aggregates responses from multiple LLM agents, appends them to the original query and feeds the enriched input to the next layer. This facilitates multi-agent synergy, enabling models to complement each other and refine predictions collaboratively. While MoA has shown that integrating multiple LLMs can be advantageous over a single model, it still faces several limitations that hinder its scalability and effectiveness:

**Which agents?** As shown in Figure 1 (b), following the scaling law of diverse LLM agents, selecting a relevant subset from a large pool of agents handling complex and diverse queries (Figure 1

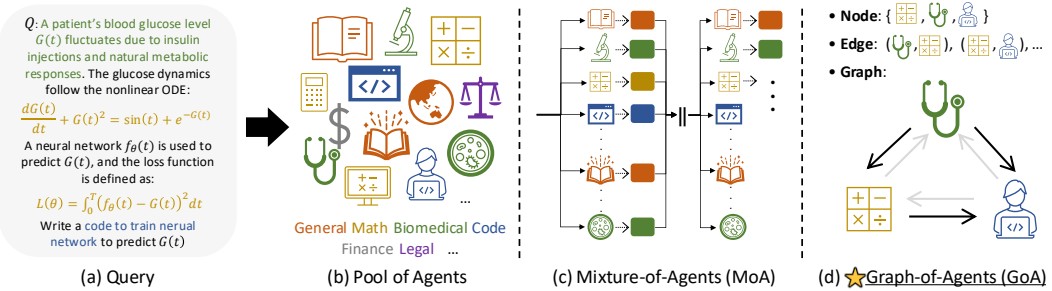

Figure 1: Current multi-agent LLM pipeline and our proposed approach.(a) Given a query that spans diverse domains (e.g., biomedical + math + code), (b) selecting and organizing agents from a large pool of LLMs to form an effective multi-agent system remains a significant challenge. (c) The current Mixture-of -Agents (MoA) approach integrates all available agents, aggregates their responses, and forwards the combined output to the next layer. However, it lacks generalizability to larger agent pools (e.g., 10 or 100) and suffers from heavy intra-layer communication overhead. (d) In contrast, our proposed Graph-of Agents (GoA) addresses this challenge through a graph-based structure. In GoA, only a subset of relevant agents is selected to form the graph's nodes, with intra-layer communication facilitated via directed message passing—flowing from more relevant agents to less relevant ones. By leveraging this graph structure, GoA achieves greater scalability and enables more efficient communication between agents, resulting in a more powerful and adaptive framework.

(a)) is a crucial challenge. The current MoA approach lacks an effective agent selection mechanism, instead forwarding queries to all available agents, leading to excessive computational costs. This not only risks multi-agent system explosion but also introduces noise from irrelevant agents, highlighting the need for a more efficient and practical agent selection strategy.

**How do they communicate?** Once agents are selected, facilitating effective communication among them becomes a pivotal challenge. MoA adopts a many-to-one aggregation scheme, but this approach has notable drawbacks: it requires collecting responses from all available agents and treating them as a single chunk, which fails to capture fine-grained interactions—such as one-to-one communication between individual agent pairs. Moreover, since different agents have varying strengths and relevance depending on the query, treating all agent messages equally can hinder consensus. Instead, adaptively weighting responses based on their relevance is crucial for improving decision-making.

**How to integrate?** Finally, when making the final decision, MoA aggregates responses by concatenating tokens from all agents. However, this approach incurs a huge computational cost, with a complexity of $\mathcal{O}(LNd)$, where $L$ is the number of communication layers, $N$ is the number of agents, and $d$ is the token length per agent. Given that token usage is directly tied to cost, this method becomes prohibitively expensive. Moreover, not all agents contribute equally valuable responses—some domain-specialized agents produce significantly higher-quality outputs than others. A scalable integration mechanism that prioritizes more reliable agents while reducing the impact of less relevant ones is essential for cost-efficient and effective decision-making.

⋆ **Our Approach.** To address these challenges, we propose a *graph-based multi-agent framework*, Graph-of-Agents (GoA), as a novel remedy to enhance multi-agent communication. GoAfundamentally rethinks multi-agent collaboration by modeling agents as nodes and their relevance-based relationships as edges, enabling structured message passing. This graph-based design allows for the selective activation of only relevant agents (as a subgraph of entire pool) while capturing inter-agent interactions through message passing and finalizing decisions via graph-based pooling. GoA follows a structured process, as shown in Figure 1 (d):

❶ **Which agents?** → *Node Sampling*: GoA begins by selecting relevant agents based on available metadata (*e.g.*, domain, task) from model cards. This information is provided to a meta-LLM, simply a general-domain LLM, which identifies the most relevant agents given the query.

❷ **How do they communicate?** → *Edge Sampling & Message Passing*: Once nodes (agents) are selected, we obtain their initial responses and ask each agent to rank others, capturing the significance of each agent's output. Based on these rankings, we construct directed edges in two perspectives: (i) Source-to-Target: Higher-ranked agents (*i.e.*, highly influential nodes with more relevant responses) propagate their information to lower-ranked agents, allowing them to refine their responses based on more confident initial answers. (ii) Target-to-Source: After lower-ranked agents

update their responses, the refined information is passed back to the higher-ranked agents, enabling them to further adjust their outputs based on the improved responses from their neighborhood agents.

❸ **How to integrate?** → *Graph Pooling*: With refined responses, GoA applies max or mean pooling, akin to graph pooling, to adaptively aggregate the outputs of multiple agents. GoA is extensively evaluated on diverse multi-domain benchmarks (MMLU, MMLU-Pro, GPQA) and domain-specific benchmarks (MATH, HumanEval, MedMCQA), demonstrating its generalizability and adaptability across a wide range of tasks. These results highlight the effectiveness of viewing multi-agent collaboration through the lens of graph structures.

It is important to note that, unlike traditional multi-agent learning method that require model fine-tuning or additional training, our proposed GoA framework, as like MoA, operates purely through the prompt interface. This design ensures compatibility with black-box LLM APIs while maintaining high adaptability across diverse domains during test-time inference.

Our contributions are three-fold:

- We identify key challenges in current multi-agent LLM systems: selecting which agents to sample, facilitating effective communication, and integrating responses efficiently.

- We formulate multi-agent collaboration as a graph-based framework and introduce GoA, incorporating node sampling, edge sampling, message passing, and graph pooling. This enables construction of a scalable multi-agent LLM ecosystem, while enhancing inter-agent communication.

- We demonstrate the effectiveness of GoA across diverse benchmarks, including MMLU, MMLU-Pro, GPQA, MATH, HumanEval, and MedMCQA. Notably, using only 3 agents, GoA outperforms recent multi-agent baselines that rely on pools of 6 agents, highlighting how graph-based structures improve both the scalability and effectiveness of multi-agent collaboration.

## 2 RELATED WORK

**LLM Reasoning and Test-Time Inference.**  Test-time reasoning with LLMs has seen rapid advances through prompt engineering techniques such as Chain-of-Thought (CoT) (Wei et al., 2022b), Tree-of-Thought (Yao et al., 2023), and Graph-of-Thought (Besta et al., 2024; Yao et al., 2024), which enable models to decompose complex problems into structured sub-tasks. While most prior work focuses on improving a single LLM's reasoning via internal prompting strategies (Zhou et al., 2023; Xu et al., 2024; Feng et al., 2024), a parallel direction has emerged around collaborative reasoning at inference time using multiple LLMs (Du et al., 2023a; Chan et al., 2023). In these multi-agent setups, multiple LLMs interact in test-time without additional finetuning, often aiming to boost factual accuracy or reasoning diversity. However, these approaches typically rely on simplistic communication protocols such as symmetric debate or sequential refinement, lacking structured mechanisms for message routing or adaptive collaboration. Our work builds on this foundation by proposing a graph-based test-time collaboration framework that formalizes agent interactions through directional message passing, enabling richer and more scalable multi-agent reasoning.

**LLM Ensembles and Multi-Agent LLM Collaboration.**  Another approach to leveraging multiple LLMs is ensemble-based inference, where outputs from several models are aggregated or selected (Fang et al., 2024; Yuan et al., 2023). Recent works introduce router mechanisms (Wang et al., 2023b; Hari & Thomson, 2023; Wang et al., 2024b; Yue et al., 2025) to reduce computational cost by selectively querying a subset of models. However, many of these ensemble strategies treat LLMs as interchangeable units without modeling their relationships. Meanwhile, graph-based structures have begun to emerge to coordinate multi-agent systems, notably in frameworks like MacNet (Qian et al., 2024) and GPTSwarm (Zhuge et al., 2024), which model agents as nodes in static DAGs. Also DyLAN (Liu et al., 2024b) performs dynamic agent activation using an Agent Importance Score computed via forward–backward peer-rating propagation and coordinates agents through a temporal feed-forward communication network. Yet, these approaches rely on predefined topologies or assume a single agent role-playing multiple personas, limiting flexibility and expressiveness, and sometimes requiring additional optimization. In contrast, our proposed framework, GoA, constructs dynamic graphs based on task relevance, enabling one-to-one communication across a diverse pool of specialized LLMs. Through node sampling, directional edge construction, and graph pooling, GoA supports efficient and adaptive collaboration—addressing the underexplored challenge of coordinating heterogeneous agents for domain-diverse reasoning at test time.

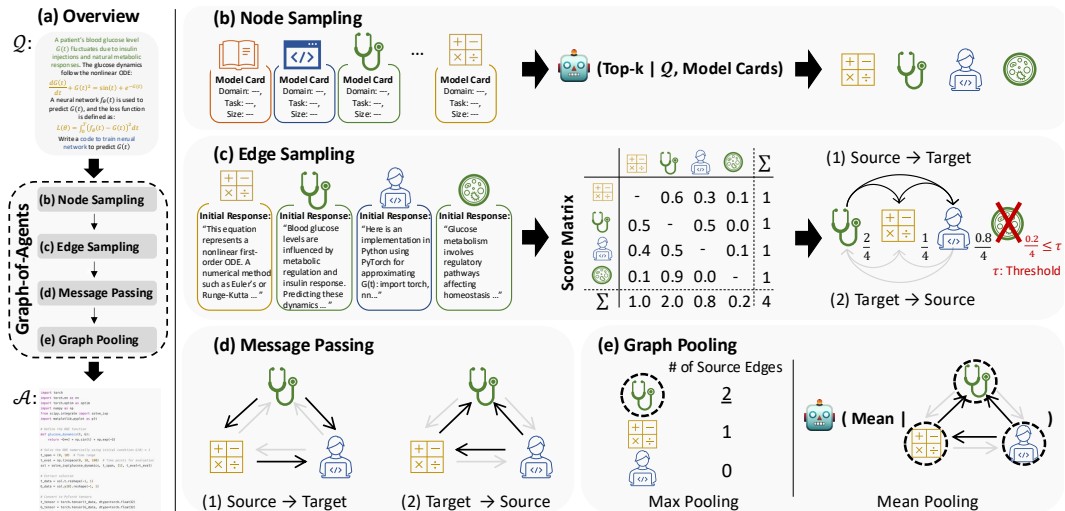

Figure 2: Overall pipeline of GoA. (a) **Overview**: Given a query ($\mathcal{Q}$) spanning diverse domains, GoA approaches multi-agent LLMs through the lens of a graph framework and produces an answer ($\mathcal{A}$). (b) **Node Sampling**: Each agent is mapped to a model card containing domain and task information. The Meta-LLM, a general-purpose LLM, takes $\mathcal{Q}$ and the model cards as input and selects the most relevant agents, forming an adaptive multi-agent framework. (c) **Edge Sampling**: After collecting initial responses from selected agents, each agent evaluates the relevance of others (excluding itself) to generate a normalized score matrix. Edges are established in Source-to-Target and Target-to-Source directions, while low-relevance nodes are pruned using a threshold $\tau$=0.05. (d) **Message Passing**: GoA first passes messages from source to target nodes, allowing lower-ranked agents to refine responses, then reverses the flow to update source nodes. (e) **Graph Pooling**: With updated responses structured as a graph, GoA outputs the final prediction via max or mean pooling.

# 3 METHODOLOGY

## 3.1 PRELIMINARIES AND NOTATIONS

**Multi-agent LLMs.** We define a system of $N$ LLM agents, where each agent $i \in 1, \ldots, N$ specializes in a particular domain or task. The agents collaboratively process a given query $\mathcal{Q}$ to generate an optimized response $\mathcal{A}$.

**As a graph.** We represent the multi-agent system as a directed graph, $\mathcal{G} = (\mathcal{V}, \mathcal{E})$, where $\mathcal{V} = v_1, \ldots, v_N$ denotes the set of agent nodes and $\mathcal{E} \subseteq \mathcal{V} \times \mathcal{V}$ is the directed edges. Once a subset of $S$ relevant agents are selected from the multi-agent pool of size $N$, the adjacency matrix $\mathbf{A} \in \mathbb{R}^{S \times S}$ is constructed. The matrix is defined such that $\mathbf{A}_{ij} > 0$ if $(v_i, v_j) \in \mathcal{E}$ and $\mathbf{A}_{ij} = 0$ otherwise.

## 3.2 OUR APPROACH: GOA

In this section, we present GoA, a novel approach specifically designed to enhance multi-agent communication through a graph-based framework, as shown in Figure 2.

**Overview.** GoA begins with node sampling, where a meta-LLM (a general-purpose LLM) selects the most relevant agents from a given pool based on the query and a model card dictionary. Each model card summarizes information extracted from the Hugging Face README file, including the LLM's domain and specialized tasks (Sec. 3.2.1). After collecting initial responses from the selected agents, we rank each agent based on scores assigned by other agents. Agents are then sorted by these scores to define two node types: source nodes (highly relevant and influential) and target nodes (less influential). This establishes bidirectional relationships: source-to-target and target-to-source (Sec. 3.2.2). In the message-passing, we first propagate messages from source to target, allowing target nodes to refine their responses based on input from more relevant agents. We then reverse the flow—target to source—so that source nodes can further refine their outputs using the updated responses. This two-step process ensures that each agent's contribution is weighted by its relevance to the query (Sec. 3.2.3). Finally, we apply graph pooling: either max pooling based on the most connected source node or mean pooling guided by the meta-LLM to generate the final answer (Sec. 3.2.4).

### 3.2.1 NODE SAMPLING

Given a query $\mathcal{Q}$, our goal is to select a subset of agents most relevant to the task. To achieve this, we leverage publicly available model cards from Hugging Face (Jain, 2022), which contain useful metadata such as the dataset the LLM was trained on, its specialized domain, model size. Using this information, we summarize each agent's model card into three key categories: (1) The domain of the LLM, (2) The specialized task, and (3) The model size and special features. Once the summarized model card is obtained, we prompt the Meta-LLM[1] to determine which agents are most likely to generate the most effective response given the query. Formally, the selected subset of agents, $\mathcal{V}_s \subseteq \mathcal{V}$, is obtained as:

$$\mathcal{V}_s = \text{Meta-LLM}(\text{Top-}k|\mathcal{Q}, \text{Model Cards}), \tag{1}$$

where Top-$k$ selects the $k$ most relevant agents based on their alignment with the query and model card information. For example, as illustrated in Figure 2 (b), if the query pertains to biomedical, mathematics, and code-related domains, the selected agents would primarily belong to these specialized areas to maximize multi-agent synergy. This approach effectively filters out unnecessary agents (e.g., law-related models), preventing agent explosion (i.e., involving an unmanageable number of agents) while maintaining relevance agents for handling the query, answering "**Which agents?**".

### 3.2.2 EDGE SAMPLING

Once the subset of selected agents $\mathcal{V}_s$ is obtained, we prompt each agent to generate its initial response to the query $\mathcal{Q}$, forming a response set $\mathcal{R} = \{v_1(\mathcal{Q}), \ldots, v_S(\mathcal{Q})\}$, where $S = |\mathcal{V}_s|$. To model inter-agent relevance, we construct a score matrix in which each agent scores the responses of all other agents (excluding its own to reduce self-bias) based on alignment with $\mathcal{Q}$. These scores are normalized such that each agent distributes a total score of 1.0 across the remaining $S-1$ agents. We then compute a relevance score $\mathcal{S}_j$ for each agent $j$ by summing the scores it receives from others:

$$\mathcal{S}_j = \sum_{\substack{i=1 \\ i \neq j}}^{S} \text{Score}_{i \rightarrow j}. \tag{2}$$

The relevance scores $\mathcal{S}$ are then used to rank agents and determine their communication roles (e.g., source vs. target). To avoid including weak or noisy responders—particularly in cases where model cards may lack detailed information—we introduce a threshold hyperparameter $\tau$. Agents with $\mathcal{S}_j < \tau$ are pruned from the communication graph, ensuring scalability and better task-fit in the constructed structure.

Using the remaining high-relevance agents, we define a weighted directed adjacency matrix $\mathbf{A} \in \mathbb{R}^{S \times S}$ to govern message passing. Each entry $\mathbf{A}_{ji}$ represents how much influence agent $i$ exerts on agent $j$ when passing messages. It is computed by normalizing the total relevance scores of all neighbors of agent $j$:

$$\mathbf{A}_{ji} = \frac{\mathcal{S}_i}{\sum_{k \in \mathcal{N}_j} \mathcal{S}_k}, \quad \text{where } \mathcal{N}_j = \{i \mid (i \rightarrow j) \in E\}. \tag{3}$$

This scoring formulation ensures that more relevant agents have proportionally greater influence, and it enables fine-grained 1-to-1 communication tailored to task-specific needs. Figure 2(c) illustrates how the score matrix and pruning mechanism dynamically shape the communication graph.

### 3.2.3 MESSAGE PASSING

Now, given the edge information and weighted adjacency matrix, we proceed with message passing, a key advantage of the graph structure. To incorporate the significance of each agent, GoA performs message passing in two steps: **Source-to-Target** followed by **Target-to-Source**.

---

[1]Any general-purpose LLM. In our experiments, we used Qwen2.5-7B-Instruct (Team, 2024) from 7–8B agent pool.

**Source-to-Target.** For highly influential nodes, it is crucial to maintain their initial strength rather than being influenced by less significant or noisy nodes. Conversely, less significant nodes benefit from receiving messages (i.e., more relevant responses to the query) from stronger nodes. Thus, we first propagate information from source nodes (higher-ranked agents) to target nodes (lower-ranked agents), allowing the latter to refine their responses based on more confident initial answers:

$$\mathcal{R}_j^{'} = v_j \left( \|_{j=i+1}^{S} \mathbf{A}_{ij} \mathcal{R}_i^{\text{sorted}} \right), \quad \text{where} \ \ i < j \le S, \tag{4}$$

where $\mathcal{R}_j^{'}$ represents the updated response for target node $j$ after receiving messages from source node $i$. Here, $v_j(\cdot)$ denotes the forward pass of LLM $j$, $\|$ represents the concatenation of neighboring responses, and $\mathcal{R}_i^{\text{sorted}}$ represents the sorted responses, ranked from highly relevant to less relevant based on the relevance scores $\mathcal{S}$ obtained in Equation 2. With these updated responses, we now proceed with the Target-to-Source step.

**Target-to-Source.** Since the previous step only updates target nodes, we also allow source nodes to refine their responses based on the improved outputs of their neighbors ($\mathcal{R}_j^{'}$), rather than the initial responses from target nodes. This enables source nodes to incorporate the consensus of their neighboring agents, indirectly influenced by the original source node, leading to further refinement:

$$\mathcal{R}_i^{''} = v_i \left( \|_{j=i+1}^{S} \mathbf{A}_{ji} \mathcal{R}_j^{'} \right), \quad \text{where} \ \ i < j \le S, \tag{5}$$

where $\mathcal{R}_i^{''}$ represents the refined response for source nodes. With both source and target nodes refining their responses collaboratively through the graph structure, we effectively address the question of "**How do they communicate?**". We now move on to the final step: response integration.

### 3.2.4 GRAPH POOLING

To address the question of "**How to integrate?**" while minimizing the computational cost associated with token stacking, we formalize response integration through graph pooling, a common approach in graph-based tasks that requires aggregating node representations into a single graph representation. Motivated by this, we propose two pooling strategies: ❶ *Max-Pooling*, which relies on the most influential node (i.e., the agent with the highest number of incoming edges, indicating a higher relevance score). ❷ *Mean-Pooling*, which balances contributions by considering responses from all selected agents but on a reduced scale, unlike MoA, which involves all available agents. Formally, this can be expressed as:

$$\mathcal{A} = \begin{cases} \mathcal{R}_{\text{max-source}}^{''} & \text{if max-pooling} \\ \text{Meta-LLM(Average}|\mathcal{R}^{''}) & \text{if mean-pooling,} \end{cases} \tag{6}$$

where $\mathcal{R}_{\text{max-source}}^{''}$ denotes the refined response of the agent with the highest number of source edges (i.e., the most relevant agent). In summary, max-pooling prioritizes the response of the most significant agent, while mean-pooling incorporates responses from all selected agents, requiring an additional forward pass through the Meta-LLM. We introduce these two variants as $\mathsf{GoA}_{\text{max}}$ and $\mathsf{GoA}_{\text{mean}}$, which will be analyzed in the experiment sections. The averaging is performed in a weighted manner using the relevance scores assigned during edge sampling.

### 3.3 GOA GENERALIZES MOA

Lastly, we demonstrate that GoA generalizes the existing MoA framework. Specifically, as illustrated in Figure 1 (c), the message-passing and response-updating procedure of MoA (Wang et al., 2024a) can be formulated as:

$$\mathcal{R}_i^{'} = v_i(\|_{j=1}^{N} \mathcal{R}_j + \mathcal{Q}). \tag{7}$$

Comparing this to Equation 4, we establish the following:

Table 1: Benchmark performance across multi-domain and domain-specific benchmarks. *Single-agent baselines*: **General**: Qwen2.5-7B-Instruct (Team, 2024), **Code**: Qwen2.5-Coder-7B-Instruct (Hui et al., 2024), **Math**: Mathstral-7B-v0.1 (AI, 2024), **Biomedical**: Bio-Medical-Llama-3-8B (Con, 2024), **Finance**: finance-Llama3-8B (Cheng et al., 2024), **Legal**: Saul-7B-Instruct-v1 (Colombo et al., 2024). *Multi-agent baselines*: **Debate** (Du et al., 2023b), **Self-Consistency (SC)** (Wang et al., 2023a), **Refine** (Madaan et al., 2023), **ReConcile** (Chen et al., 2023), **MoA** (Wang et al., 2024a), and **Self-MoA** (Li et al., 2025). Our proposed framework, GoA **(Graph-of-Agents)**, despite using only *3 agents* (i.e., top-$k$=3), outperforms both multi-agent baselines with *6 agents* and single-agent models, demonstrating strong collaborative synergy and capability across both multi-domain and domain-specific tasks. All performance is measured using zero-shot CoT in test-time.

| | | Multi-Domain | | | Domain-Specific | | |
| --- | --- | --- | --- | --- | --- | --- | --- |
| | | MMLU | MMLU-Pro | GPQA | MATH | Human Eval | MedMCQA |
| **Single-Agent** | General | 77.61 | 53.90 | 32.83 | 69.00 | 81.50 | 55.22 |
| | Code | 68.04 | 42.33 | 33.84 | 59.60 | **85.37** | 45.57 |
| | Math | 63.47 | 37.19 | 30.81 | 48.60 | 57.93 | 45.25 |
| | Biomedical | 46.60 | 27.90 | 25.25 | 18.80 | 20.73 | 47.00 |
| | Finance | 54.11 | 25.52 | 28.28 | 13.80 | 27.44 | 42.08 |
| | Legal | 55.86 | 27.57 | 30.30 | 12.40 | 36.59 | 41.50 |
| **Multi-Agent** *(6 Agents)* | Debate | 72.53 | 47.05 | 29.29 | 69.60 | 40.24 | 53.05 |
| | SC | 77.97 | 54.12 | 36.36 | 69.80 | 82.57 | 55.70 |
| | Refine | 77.40 | 54.71 | 38.92 | 71.60 | 80.49 | 54.94 |
| | ReConcile | 69.61 | 44.19 | 34.34 | 45.60 | 50.20 | 54.60 |
| | MoA | 75.71 | 53.33 | 32.83 | 65.80 | 76.22 | 54.94 |
| | Self-MoA | 78.14 | 54.19 | 33.84 | 68.20 | 79.27 | 55.56 |
| **Multi-Agent** *(3 Agents)* | GoA$_{Max}$ | **79.18** | **54.78** | 39.98 | 69.83 | 84.67 | **60.04** |
| | GoA$_{Mean}$ | 78.52 | 54.27 | **40.54** | **73.12** | 84.98 | 57.92 |

**Proposition 1** *Graph-of-Agents (GoA) reduces to MoA when the node sampling parameter $k$ equals the total number of agents $N$, the adjacency matrix is fully connected with all edge weights set to 1, i.e., $\mathbf{A} \in \mathbb{R}^{N \times N} = 1$, and a self-loop is included with the initial query ($\mathcal{Q}$) at each layer, ultimately aggregated via mean pooling.*

Thus, GoA serves as a more flexible and extensible generalization of multi-agent communication frameworks. By introducing structured message passing, adaptive agent selection, and weighted aggregation, GoA can scale to large agent pools while maintaining efficiency and robustness. As our work focuses on test-time inference, the prompts used in this study are provided in Appendix B.

## 4 EXPERIMENTS

### 4.1 IMPLEMENTATION DETAILS

**Benchmarks.** We evaluate performance on two multi-domain (MMLU (Hendrycks et al., 2020), MMLU-Pro (Wang et al., 2024c), GPQA (Rein et al., 2023)) and three domain-specific benchmarks (MATH (Lightman et al., 2023), HumanEval (Chen et al., 2021), MedMCQA (Pal et al., 2022)). For MMLU and MMLU-Pro, due to their large sizes, we used stratified sampling: 50 samples per category across 57 categories for MMLU, and 150 samples per category across 14 categories for MMLU-Pro. For the agent pool, we primarily leveraged six LLMs with 7–8B parameters, covering diverse domains such as General, Code, Math, Biomedical, Finance, and Legal. The specific LLMs are listed in Table 1.

### 4.2 MAIN RESULTS

**Effectiveness.** In Table 1, we present benchmark results across both multi-domain and domain-specific tasks, comparing single-agent baselines, multi-agent baselines with 6 agents, and our proposed GoA (3 agents). ❶ Among single-agent baselines, the general-purpose model achieves the highest average performance (61.15), but falls short of the best-performing multi-agent models. Specialized agents (e.g., Math, Biomedical, Finance) tend to perform well only in narrow domains

and underperform in others, leading to lower overall averages. ❷ Multi-agent baselines with 6 agents—especially Refine (62.15) and Self-MoA (61.63)—demonstrate improved performance over all single-agent baselines, confirming the benefits of agent collaboration. However, their effectiveness varies across benchmarks, with no single method dominating all tasks. ❸ Our proposed GoA method outperforms all baselines across most metrics. $GoA_{\text{Max}}$ achieves the highest average score, with top performance on MMLU (79.18), MMLU-Pro (54.78), and MedMCQA (60.04). $GoA_{\text{Mean}}$ records the best scores on GPQA (40.54), MATH (73.12), and HumanEval (84.98), showing robust and consistent gains across both reasoning and domain-specific tasks. ❹ Notably, while other multi-agent methods rely on integrating all 6 agents, GoA achieves superior results using only 3, suggesting that selective and structured agent collaboration can be more effective than full integration.

**Efficiency.** Another crucial perspective for test-time LLM collaboration is efficiency—that is, handling multiple LLMs in a scalable manner. Table 2 presents a comparison between MoA and GoA on the MMLU-Pro dataset in terms of accuracy, number of LLM calls, average token usage (input + output), and latency. Unlike MoA, which employs multiple

Table 2: Efficiency analysis in MMLU-Pro.

|  | Acc. | Calls | Tokens (k) | Time (s) |
|---|---|---|---|---|
| MoA | 53.33 | 19 | 56.05 | 240.26 |
| $GoA_{\text{Max}}$ | 54.78 | 11 | 19.18 | 100.43 |
| $GoA_{\text{Mean}}$ | 54.27 | 12 | 22.58 | 118.52 |

rounds of proposers (i.e., all available agents) and a final aggregator, GoA reduces LLM usage and latency while improving accuracy. This improvement stems from its design philosophy—node and edge sampling followed by graph pooling—which selectively involves only relevant agents in the system, making it a more efficient and practical approach for multi-agent LLM collaboration.

**Scaling Up.** To evaluate generalization to proprietary models, we tested GoA with gpt-4o on GPQA, MedMCQA (100 sampled), and HumanEval. Table 3 shows multi-agent setups outperform the single-agent baseline. Notably, while DyLAN (Liu et al., 2024a) employs eight specialized agents (e.g., 'Python Assistant', 'Algorithm Developer'), our GoA with only three agents achieves higher performance, mirroring trends seen with open-source models. This advantage stems from our graph-based reasoning and relevance-aware message-

Table 3: Scaling up with gpt-4o model.

|  | GPQA | MedMCQA | HumanEval |
|---|---|---|---|
| gpt-4o | 47.47 | 77.00 | 90.20 |
| Debate (6 Agents) | 53.03 | 80.00 | 85.98 |
| SC (6 Agents) | 54.27 | 81.00 | 92.07 |
| Refine (6 Agents) | 54.98 | 82.00 | 91.92 |
| Reconcile (6 Agents) | 53.03 | 75.00 | 91.46 |
| MoA (6 Agents) | 50.51 | 80.00 | 92.07 |
| DyLAN (8 Agents) | **58.89** | 81.00 | 92.07 |
| $GoA_{\text{Max}}$ (3 Agents) | 55.05 | 82.00 | 93.29 |
| $GoA_{\text{Max}}$ (6 Agents) | 56.57 | **83.00** | **93.90** |

passing mechanism (Section 4.3), which enables targeted and noise-resilient communication. These results highlight that well-designed communication strategies can be more effective than simply increasing the number of agents, underscoring both the effectiveness and scalability of GoA.

## 4.3 WHY GRAPH?

The key aspect of GoA is its graph-based framework for multi-agent LLM collaboration. In this section, we show how the **graph structure** and the **new message-passing** benefit LLM collaboration.

**Graph-based Reasoning.** Figure 3 presents a case study illustrating how a graph-based framework improves reasoning by comparing recent MoA with our proposed GoA when handling an anatomy-domain from the MMLU dataset. In MoA, all available agents are used, including those from unrelated domains such as math and code. These irrelevant agents introduce noise (e.g., 'Answer: 1'), which negatively affects the final prediction. In contrast, GoA avoids irrelevant agent usage by applying node sampling followed by edge sampling, thereby constructing a query-specific graph structure. This enables targeted message-passing among relevant agents, leading to more accurate discussions (e.g., 'Answer: 0') and final predictions. Overall, this comparison shows how a graph-based framework enhances reasoning in multi-agent scenarios by leveraging structured interactions and relevance-aware message passing.

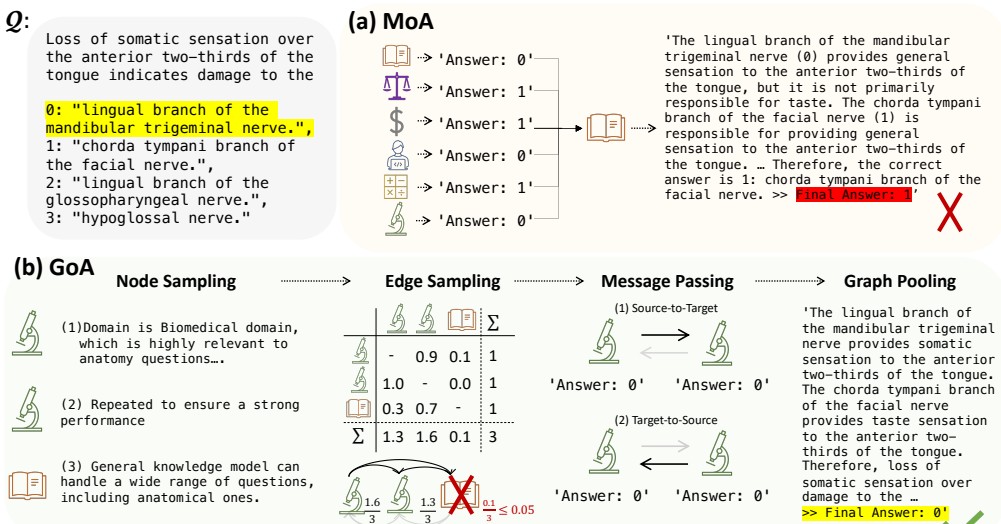

Figure 3: Reasoning process of (a) Mixture-of-Agents (MoA) and (b) Graph-of-Agents (GoA).

**Relevance-aware Message-Passing.** We now delve into GoA's relevance-aware message-passing framework under a stricter fixed-node setting, targeting the HumanEval benchmark with three code-specific models. As shown in Table 4, the baseline models exhibit varying performance—Qwen2.5-Coder-7B-Instruct performs best, while the others fall behind. In the multi-agent setting, we compare our approach against existing frameworks: MoA

Table 4: Effectiveness of Message-Passing.

|  | HumanEval |
|---|---|
| Qwen2.5-Coder-7B-Instruct | 85.37 |
| Seed-Coder-8B-Instruct | 80.49 |
| deepseek-coder-7b-instruct-v1.5 | 73.17 |
| MoA - MoE-based (3 Agents) | 85.37 |
| Debate - Debate-based (3 Agents) | 71.95 |
| Reconcile - Confidence-based (3 Agents) | 80.61 |
| GoA$_{Max}$ - Graph-based (3 Agents) | **85.98** |

(MoE-based) (Wang et al., 2024a), Debate (Du et al., 2023b), and Reconcile (confidence-based) (Chen et al., 2023), with our GoA (graph-based) achieving the highest performance. This gain stems from its tailored message-passing mechanism, where edges are formed based on node relevance to the given question: information flows from highly relevant nodes (e.g., Qwen2.5-Coder-7B-Instruct) to less relevant ones (e.g., deepseek-coder-7b-instruct-v1.5), which update their internal state, followed by a feedback phase that produces an aggregated response. These results demonstrate that graph-based approaches—especially when combined with relevance-aware message passing—offer a promising paradigm for LLM collaboration.

## 4.4 ABLATION STUDY

Table 5 presents an ablation analysis on MMLU-Pro and GPQA. The top row shows the original GoA setting with the best performance. ❶ Interestingly, reversing the message-passing direction causes the largest performance drop (–2.60 on MMLU-Pro and –5.05 on GPQA), even worse than removing only one direction (Source-to-Target or Target-to-Source). This matches our expectation: when less-influential nodes become artificially dominant, they disrupt the intended flow and inject noise back into the original source nodes. This underscores a

Table 5: Ablation study on modules, top-$k$, and $\tau$.

|  | MMLU-Pro | GPQA |
|---|---|---|
| GoA (Top-$k$=3, $\tau$=0.05) | **54.78** | **39.98** |
| Reverse Message Passing | 52.18 | 34.93 |
| w/o Target-to-Source | 53.66 | 38.03 |
| w/o Source-to-Target | 52.21 | 36.12 |
| w/o Scoring ($\mathbf{A}_{ij}=1$) | 52.91 | 37.34 |
| Top-$k$=2 | 53.54 | 36.75 |
| Top-$k$=5 | 54.65 | 39.13 |
| $\tau$=0.1 | 53.12 | 38.43 |
| $\tau$=0.2 | 52.78 | 37.12 |

key design insight—preserving the correct bidirectional flow, (1) Source-to-Target and (2) Target-to-Source, is essential for GoA's effectiveness.. Removing Target-to-Source message passing leads to a noticeable drop (–1.12 MMLU-Pro, –1.95 GPQA), showing that feedback from target nodes is crucial. ❷ Removing Source-to-Target causes even larger degradation (–2.57, –3.86), highlighting the importance of initial information flow. Together, these results confirm the complementary

role of bidirectional message passing. ❸ Disabling edge scoring ($A_{ij} = 1$) consistently reduces performance, validating the benefit of relevance-based weighting. ❹ Varying the number of agents shows $k = 2$ limits diversity while $k = 5$ introduces slight degradation (likely requiring tailored $\tau$), with $k = 3$ remaining scalable and performant. ❺ Adjusting the edge threshold $\tau$ shows that overly sparse graphs ($\tau = 0.1, 0.2$) harm performance, while $\tau = 0.05$ balances focus and connectivity. Overall, these results demonstrate that bidirectional message passing, adaptive edge scoring, and selective agent activation all contribute to GoA's effectiveness.

## 5 CONCLUSION

In this study, we introduce GoA, a graph-based framework that re-designs multi-agent communication to address key challenges: selecting relevant agents, enabling effective communication, and integrating responses efficiently. Extensive experiments demonstrate consistent gains across diverse benchmarks. We believe GoA can advance the development of large-scale collective intelligence in the emerging multi-agent LLM era.

## ACKNOWLEDGEMENTS

This research was partially funded by the National Institutes of Health (NIH) under award 1R01EB03710101. The views and conclusions contained in this document are those of the authors and should not be interpreted as representing the official policies, either expressed or implied, of the NIH. This research was also partially supported by the Amazon Research Award.

## ETHICS STATEMENT

We propose a graph-based multi-agent LLM collaboration framework, GoA. However, the LLMs used in GoA may still exhibit inherent biases and undesirable traits from pretraining. Consequently, its outputs carry similar risks of misuse as other test-time methods.

## REPRODUCIBILITY STATEMENT

We provide our code at the link: https://github.com/UNITES-Lab/GoA. All experiments were conducted on A6000 GPU. The datasets used are publicly available.

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

# APPENDIX

## A  THE USE OF LARGE LANGUAGE MODELS (LLMS)

We use ChatGPT[2] exclusively for grammar checking and text refinement. Its role was limited to polishing author-written text and did not involve research ideation.

## B  PROMPTS

As our proposed GoA framework operates during test-time inference, the detailed prompts for each stage are provided below:

- **Model Card Extraction (Example: Qwen2.5-Coder-7B):**

> **Model Card Extraction - System Prompt**
>
> You are an expert in analyzing and summarizing AI model documentation.

---

[2] https://chatgpt.com/

---

**Model Card Extraction - User Prompt**

You are given the README file of a language model:

```
language:
- en
base_model:
- Qwen/Qwen2.5-Coder-7B
pipeline_tag: text-generation
library_name: transformers
tags:
- code
- codeqwen
- chat
- qwen
- qwen-coder
---

# Introduction
Qwen2.5-Coder is the latest series of Code-Specific ...
- code generation, code reasoning and code fixing.
- ...
```

Please extract and summarize the model's key characteristics clearly and concisely in the following structured format:

1. **Domain**: The primary domain or application area the model is designed for (e.g., general-purpose, biomedical, finance, coding, math, etc.).
2. **Task Specialization**: Describe the task types the model is designed for or excels at. Be as specific as possible, including the domain context of each task (e.g., biomedical question answering, clinical decision support, financial sentiment classification, code generation). Do not include performance metrics, benchmark names, or evaluation results.
3. **Parameter Size**: The number of parameters in the model (approximate if not explicitly stated).
4. **Special Features**: Any distinguishing aspects such as fine-tuning datasets (if applicable).

Your summary will later be used to compare multiple models for selection purposes. Return your answer in bullet-point format, using the exact field names shown above. Keep it concise but specific enough for model comparison.

Answer:

---

- **Initial Response Generation**:

  Each sampled model generates an initial response in a structured JSON format to ensure reliable answer extraction. The following instruction is appended to the original question:

  > **Initial Response Generation — User Prompt**
  >
  > `{instruction}`
  >
  > Provide brief reasoning (2-3 key sentences), then output your final answer in JSON format:
  >
  > `{``reasoning``:  ``<brief reasoning>``, ``answer``: ``<answer_format_hint>``, ``confidence_level``:  ``<a float between 0.0 and 1.0>``}`
  >
  > Please strictly output in JSON format.

  where `answer_format_hint` is task-specific: *"a mathematical expression or number"* for MATH/AIME, *"the completed Python function code"* for HumanEval, or *"one of the answer choices: A, B, C, D (etc)"* for multiple-choice tasks. Here, we introduce a confidence level to filter out models with limited capability to produce outputs in the desired format; in such cases, we simply replace them with a general-domain model.

- **Node Sampling**:

  All prompts in the pipeline use a single user-role message (no separate system prompt) for simplicity and broader model compatibility. The node-sampling prompt selects the top-$k$ models based on model descriptions:

  > **Node Sampling — User Prompt**
  >
  > Select `{top_k}` models best suited for this question.
  >
  > Question: `{messages[0]}`
  >
  > Available models: `{model_descriptions}`
  >
  > Selection criteria (in priority order):
  >
  > 1. Domain match — prefer models trained in the question's domain
  > 2. Task specialization — prefer models fine-tuned for the required skill
  > 3. Include at least one generalist model if applicable
  > 4. Prefer larger models only when the size gap is significant
  >
  > Rules:
  >
  > - Output exactly `{top_k}` comma-separated indices from [0, `{max_index}`]
  > - You may repeat an index if the model is highly relevant
  > - No explanations
  >
  > Example: `{example_dict[top_k]}`
  > Answer:

- **Edge Sampling**:

  > **Edge Sampling — User Prompt**
  >
  > Score the following `{N}` model responses to this question.
  >
  > Question: `{messages[0]}`
  >
  > Responses (in order: `{other_models}`): `{other_responses}`
  >
  > Assign a score to each response based on correctness, coherence, and relevance. Scores must sum to exactly 1.0. Output only a comma-separated list of `{N}` scores.
  >
  > Example: `{example_str}`
  > Answer:

- **Message-Passing (1)** *Source-to-Target*:

  Each target model refines its response by considering the responses from its source models. Source responses are annotated with relevance labels derived from normalized edge scores: *high relevance* ($w > 0.7$), *moderate relevance* ($0.4 < w \leq 0.7$), or *low relevance* ($w \leq 0.4$).

  ---
  **Source-to-Target — User Prompt**

  Refine your answer by considering other models' responses.

  Question: {`messages[0]`}

  Your initial response: {`initial_responses[target]`}

  Other models' responses (ranked by relevance):

  *Model i (high relevance):* {`response`$_i$}
  *Model j (moderate relevance):* {`response`$_j$}
  *Model k (low relevance):* {`response`$_k$}

  Integrate useful insights from these responses to improve your answer. Be critical — some information may be incorrect.

  ---

- **Message-Passing (2)** *Target-to-Source*:

  Each source model finalizes its response after seeing how the target models refined their answers. The same relevance labeling scheme is applied.

  ---
  **Target-to-Source — User Prompt**

  Other models refined their answers after seeing yours. Use their improvements to finalize your response.

  Question: {`messages[0]`}

  Your initial response: {`initial_responses[source]`}

  Updated responses from other models:

  *Model i (high relevance):* {`response`$_i$}
  *Model j (moderate relevance):* {`response`$_j$}
  *Model k (low relevance):* {`response`$_k$}

  Write your final response, incorporating valuable refinements. Be critical — some information may be incorrect.

  ---

- **Graph-Pooling**:

  A meta-LLM synthesizes the refined responses into a single final answer.

  ---
  **Graph Pooling — User Prompt**

  Synthesize these model responses into one final answer.

  Question: {`messages[0]`}

  Model responses:
  {`input_responses`}

  Produce an accurate, coherent answer integrating the best insights. Be critical — some information may be incorrect.

  ---

