# OpenReview forum: "Graph-of-Agents: A Graph-based Framework for Multi-Agent LLM Collaboration"
_ICLR.cc/2026/Conference — ICLR 2026 Poster_

### Official Review · Reviewer_Usox · 2025-10-19

**Soundness:** 3
**Presentation:** 3
**Contribution:** 2
**Rating:** 4
**Confidence:** 4

**Summary:**

The paper proposes Graph-of-Agents (GoA), a graph-based framework for orchestrating multi-agent collaboration among Large Language Models (LLMs). GoA addresses three key limitations of existing approaches like Mixture-of-Agents (MoA): (1) inefficient agent selection, (2) unstructured communication, and (3) costly response integration. It introduces a pipeline of node sampling, edge sampling, bidirectional message passing, and graph pooling (max/mean) to produce a final answer. Evaluated on diverse benchmarks (MMLU, GPQA, MATH, etc.), GoA with only 3 agents outperforms MoA and other baselines using 6 agents, demonstrating superior efficiency and effectiveness.

**Strengths:**

1) GoA creatively reframes multi-agent LLM collaboration as a dynamic graph construction problem, combining agent metadata, relevance-based edge formation, and directional message passing. This is a synthesis of graph neural network concepts with test-time LLM orchestration, distinct from static ensembles or debate-based methods.
2) The methodology is well-structured and technically sound. The ablation study (Table 5) convincingly validates each component (e.g., bidirectional message passing, relevance scoring). Experiments span 6 diverse benchmarks and compare against 6 strong multi-agent baselines, with clear metrics (accuracy, token usage, latency).
3) The paper is clearly written and well-organized.

**Weaknesses:**

1) The proposed multi-agent collaborative framework is evaluated with only a small number of agents (e.g., 3 in Table 1), which fails to demonstrate the scalability or advantages of the method in more complex, large-scale multi-agent settings. Moreover, The ablation shows k=5 slightly underperforms k=3. Is this due to increased noise or insufficient message-passing depth? Would adding more rounds of message passing help with larger agent sets?
2) The paper lacks a clear comparison with workflow-based multi-agent systems or auto-workflow generation [1]. It remains unclear whether the proposed approach—selecting agents based on task–agent relevance—outperforms hand-crafted workflows, and the selection mechanism itself shows limited novelty.
3) Key graph-based multi-agent baselines, such as MacNet (Qian et al., 2024) and GPTSwarm (Zhuge et al., 2024), are not included in the experiments, weakening the empirical evaluation.
4) The method is not tested on established complex multi-agent benchmarks that require coordination, such as WebAgent, GUI navigation, or WebShop tasks, raising concerns about its generalizability and practical utility.
5) The results show that their model with 3 agents outperforms the baselines with 6 agents. However, the agents set for selection is larger than 3. Moreover, the performance of GoA_max and GoA_mean is unstable for various tasks. Particularly, both GoA_max and GoA_mean perform similarly with the baselines. The improvement of the proposed method is very limited.
6) The manuscript contains grammatical errors (e.g., “... and graph pooling,GoA supports efficient and ...”), which affect readability and professionalism.

[1] Jiayi Zhang, Jinyu Xiang, Zhaoyang Yu, Fengwei Teng, Xionghui Chen, Jiaqi Chen, Mingchen Zhuge,
Xin Cheng, Sirui Hong, Jinlin Wang, et al. Aflow: Automating agentic workflow generation. 2025.

**Questions:**

1) In the main experiment (Table 1), the authors use 3 agents—how many candidate agents were available for selection? How do baseline methods perform when restricted to the same 3 agents? What is GoA’s performance with 6 agents? Which base model is used in Table 1? The comparison appears unfair unless all methods use the same number and type of agents.
2) The method constructs a graph based on agents’ initial responses to determine execution order, then re-executes the task using this order. Does this two-stage process significantly increase computational or implementation complexity? Can the constructed graph capture multi-layer or hierarchical dependencies among agents?
3) Intuitively, the benefit of GoA should grow with more agents, yet Table 3 shows limited gains when increasing agent count. How do other models in Table 1 (e.g., Refine) perform under the same GPT-4o setting used in Table 3? Given that Refine already achieves strong results in Table 1, is GoA’s improvement marginal?
4) You cite router-based ensembles (e.g., Wang et al. 2023c, Hari & Thomson 2023) but don’t compare against them. How does GoA’s node sampling + graph communication compare to learned or prompt-based routing in terms of accuracy and cost?

---

> ### Author Response · Authors · 2025-11-23
> **Response to Reviewer Usox (1/4)**
>
> We thank reviewer **Usox** for acknowledging our work as `creatively reframing multi-agent collaboration as dynamic graph construction` and for recognizing the methodology as `well-structured and technically solid.` We address the raised concerns below:
>
> ---
>
> **[W1. Adding more agents]**
>
> Thanks for bringing this point up. Referring to Table 1 (top-k = 3) and GoA’s ablation study in Table 5 (top-k = 2,5), we have already varied the number of agents. We additionally include the k = 6 case below:
>
> | Agent Pool={G,C,M,B,F,L} | MMLU-Pro | GPQA  |
> |--------------------------|----------|-------|
> | top-k=2                  |    53.54 | 36.75 |
> | top-k=3 (paper)          |    **54.78** | **39.98** |
> | top-k=5                  |    54.65 | 39.13 |
> | top-k=6                  |    54.27 | 38.45 |
>
> As shown above—and as the reviewer suggested—adding more agents does not necessarily lead to performance gains, especially in a heterogeneous agent-pool spanning diverse domains. Our goal is to create an environment where agents interact synergistically, yet as seen in Table 1 (main paper), domain-specific agents alone are often individually weak (e.g., B, F, L agents score 27.90, 25.52, and 27.57 on MMLU-Pro), likely due to domain mismatch, limited coverage, or insufficient instruction-tuning. Therefore, selecting a subset of relevant agents and emphasizing more task-relevant, influential agents becomes crucial—an ability that existing approaches lack, and that GoA uniquely enables through relevance-guided sampling and graph-based communication. Consequently, top-k = 3 yields better performance than naively increasing the number of agents, since including weaker or domain-irrelevant agents introduces noise (given their inherently weak single-agent performance) rather than useful signal. This aligns with DyLAN [1], which similarly observes that a smaller set of well-chosen agents can outperform a larger unfiltered group.
>
> However, we would also like to highlight Table 3 in the main paper, where we evaluated a more controlled scaling scenario—namely, a homogeneous agent setting—by fixing the agent pool to a single LLM (gpt-4o) and simply varying the number of agent copies. The results are shown below:
>
>
> | Agent Pool={GPT-4o}              | GPQA      | MedMCQA   | HumanEval |
> | -------------------- | --------- | --------- | --------- |
> | gpt-4o               | 47.47     | 77.00     | 90.20     |
> | GoA (3 Agents)      | 55.05     | 82.00     | 93.29     |
> | GoA (6 Agents)      | **56.57** | **83.00** | **93.90** |
>
> In this controlled setting where all agents have uniform capability, GoA with 6 agents consistently outperforms GoA with 3 agents across all benchmarks. This demonstrates that when agent quality is consistent (single-LLM replication), increasing agent count can indeed improve performance.
>
> We have included this experiment in Appendix I, highlighted in blue in the updated manuscript.
>
> [1] A Dynamic LLM-Powered Agent Network for Task-Oriented Agent Collaboration

---

> ### Author Response · Authors · 2025-11-23
> **Response to Reviewer Usox (2/4)**
>
> ---
>
> **[W2 & 3. Limited related works and graph-based multi-agent baselines]**
>
> Thank you for pointing out these important references. The summary and key differences relative to GoA are as follows:
> - **AFLOW** [1] automates the construction of agentic workflows using code-structured control graphs and Monte-Carlo search. These workflows remain static across examples, with a fixed invocation sequence and logic. In contrast, GoA does not attempt to discover procedural workflows; rather, it dynamically selects agents conditioned on the current problem via node sampling and relevance-based message passing. Thus, the two settings are complementary but operate along different axes: AFLOW optimizes execution structure, while GoA optimizes adaptive agent participation and collaboration.
> - **MacNet** [2] proposes a multi-agent collaboration network using directed acyclic graphs processed through topological ordering. While GoA also employs a graph structure, MacNet relies on a predefined topology—e.g., chain, star, tree, random—whereas GoA constructs a task-specific graph dynamically based on agent relevance. Furthermore, GoA employs a unique bidirectional message-passing mechanism, emphasizing the influence of the source node (i.e., more relevant agents), enabling information flow that better preserves agent expertise and prevents noise amplification.
> - **GPTSwarm** [3] treats agents as nodes in an optimizable computation graph, learning both node-level prompt transformations and edge-level communication pathways via REINFORCE optimization. In contrast, GoA requires neither node-level nor edge-level training, and operates purely at test-time in an inference-only manner.
>
> Finally, among the above baselines, during the rebuttal period we additionally ran experiments on **MacNet** to evaluate how an existing graph-based collaboration method compares with GoA. The results are shown below:
>
> | Agent Pool={G,C,M,B,F,L} | MMLU-Pro | GPQA   |
> |--------------------------|----------|--------|
> | Refine                   |    54.71 |  38.92 |
> | ReConcile                |    44.19 |  34.34 |
> | MoA                      |    53.33 |  32.83 |
> | Self-MoA                 |    54.19 |  33.84 |
> | MacNet - Chain           |    50.67 |  36.36 |
> | MacNet - Star            |    50.52 |  35.35 |
> | MacNet - Tree            |    50.52 |  35.35 |
> | MacNet - Random          |    50.71 |  32.83 |
> | GoA_Max                  |    54.78 |  39.98 |
> | GoA_Mean                 |    54.27 | 40.554 |
>
> Here, we find that across diverse fixed topologies—chain, star, tree, and random—MacNet consistently underperforms GoA, and even falls below MoA and Self-MoA in certain cases. This indicates that in multi-agent graph design, **incorporating agent relevance and leveraging relevance-weighted bidirectional message passing** is crucial for boosting reasoning performance. Unlike static graph layouts, GoA adapts the communication structure to each problem instance, ensuring that the most relevant agents exert greater influence over the system.
>
> Additionally, as shown in Table 1 of the MacNet paper, the GPTSwarm baseline performs below MacNet. Therefore, given that GoA outperforms MacNet, it is reasonable to infer that GoA would also outperform GPTSwarm under comparable settings.
> For the final manuscript, we will expand the MacNet comparison to additional benchmarks (e.g., MATH, MedMCQA) and incorporate these results into the main table (Table 1) of the paper.
>
> We have included this experiment in Appendix H, highlighted in blue in the updated manuscript.
>
> [1] AFlow: Automating Agentic Workflow Generation
> [2] Scaling Large-Language-Model-based Multi-Agent Collaboration.
> [3] GPTSwarm: Language Agents as Optimizable Graphs.
>
> ---
>
> **[W4. Multi-agent benchmarks]**
> Thank you for the suggestion. While the current benchmarks used in our study (e.g., MMLU, GPQA, MedMCQA) are not interactive environments, our framing is **consistent with the Mixture-of-Agents (MoA) line of work**—both conceptually (multiple LLMs collaborating as an agent system, illustrated in Figure 1 of the main paper) and in evaluation setup, which similarly relies on question–answering tasks (e.g., AlpacaEval 2.0, MT-Bench, FLASK) to study test-time agent collaboration.
> That said, we appreciate the suggestion to explore more interactive, environment-grounded agentic benchmarks, such as WebAgent and GUI navigation. Incorporating GoA into such settings—where agents interact with tools, execute actions, or access the web—would require adapting the agent pool and communication protocols, and is indeed a promising direction for future work. In particular, GoA’s graph-based construction could help integrate heterogeneous tool responses into a coherent final answer, somewhat analogous to GPTSwarm’s application of Self-Consistency.
>
> We have updated this future work in Appendix E, highlighted in blue in the updated manuscript.

---

> ### Author Response · Authors · 2025-11-23
> **Response to Reviewer Usox (3/4)**
>
> ---
>
> **[W5. 3 agent Performance and GoA_max and GoA_mean]**
> In **[W1. Adding more agents]**, we demonstrated the conditions under which adding more agents can either saturate performance or lead to improvements by separating the heterogeneous vs. homogeneous settings, with experiments across 2, 3, 5, and 6 agents.
> On top of this, for **GoA_max vs. GoA_mean**, our framework intentionally draws inspiration from graph pooling schemes commonly used in Graph Neural Networks (GNNs). As such, the choice between max pooling and mean pooling should be viewed as a **hyperparameter** and a degree of freedom determined by user preference and the nature of the task.
>
> - **Max pooling** reduces the number of calls compared to mean pooling (Table 2), resulting in lower token usage and faster inference. It is particularly suitable when we want a highly influential agent (source node) to strongly guide the final prediction.
>
> - **Mean pooling**, by contrast, is preferred when the selected agents contribute complementary views and we wish to synthesize their responses into a single unbiased answer without overemphasizing any particular agent.
>
> We also refer back to Reviewer **7qjd**’s concern in **[Q1. Failure case of GoA]** [(link)](https://openreview.net/forum?id=34cANdsHKV&noteId=fmYoFbkHpx), where we illustrated cases in which mean or max pooling may fail under different conditions. This reinforces the argument that leaving the choice between max and mean pooling as a hyperparameter is beneficial, as different types of questions require different aggregation behaviors—sometimes emphasizing a dominant expert matters, other times careful integration matters.
>
> Lastly, we would like to emphasize that the key highlight of GoA is not solely performance improvement itself. As we stated in the Introduction, GoA was designed to achieve both performance and also **efficiency**. Table 1 shows that involving only **3 relevant agents** can match or even outperform existing multi-agent baselines that require **6 agents**, thereby reducing computational cost and inference overhead.
>
> ---
>
> **[W6. Grammatical Errors]**
> Thank you for pointing this out. We have corrected the identified spacing issue and highlighted the change in blue in the revised manuscript. Please let us know if you notice any additional grammatical issues that we should address.
>
> ---
>
> **[Q1. Details of 3 agent settings]**
> As the reviewer noted, in Table 1 we use 3 agents selected via top-$k$ node sampling from the agent pool. In our main setting, we simulate an agent pool of 6 LLMs spanning diverse domains: General, Code, Math, Biomedical, Finance, and Legal. We also extended this agent pool to 10 agents as requested by reviewer **4HpR** in **[Q1. Larger or more diverse agent pools]** [(link)](https://openreview.net/forum?id=34cANdsHKV&noteId=4rdVAehn6u) (see Appendix F).
>
> The base LLM for each domain is listed in the caption of Table 1, and GoA’s performance with 6 agents is discussed in **[W1. Adding more agents]**.
>
> For a more controlled setting, we refer to Table 4 in the main paper, where the agent pool is restricted to 3 code-specialized models, and only the collaboration mechanism differs across methods:
>
> | Method                          | HumanEval |
> | ------------------------------- | --------- |
> | Qwen2.5-Coder-7B-Instruct       | 85.37     |
> | Seed-Coder-8B-Instruct          | 80.49     |
> | deepseek-coder-7b-instruct-v1.5 | 73.17     |
> | Debate (3 Agents)               | 71.95     |
> | Reconcile (3 Agents)            | 80.61     |
> | MoA (3 Agents)                  | 85.37     |
> | **GoA-Max (3 Agents)**          | **85.98** |
>
> Here, we observe that GoA’s graph-based collaboration, equipped with relevance-based node and edge sampling along with bidirectional message passing, leads to improved performance compared to existing multi-agent collaboration baselines.

---

> ### Author Response · Authors · 2025-11-23
> **Response to Reviewer Usox (4/4)**
>
> ---
>
> **[Q2. Cost of execution of two-stage process and multi-layer experiment]**
> Thank you for the detailed question regarding the execution. As the reviewer noted, GoA operates with bidirectional message passing once the graph is constructed, and the complexity breakdown for this process is shown below using the HumanEval benchmark:
>
> | Human Eval | Acc   | # of Calls | # of Tokens (k) | Time   |
> |------------|-------|------------|-----------------|--------|
> | MoA        | 76.22 |         19 |           29.44 | 122.52 |
> | GoA-max    | 84.67 |         11 |           10.21 |  59.04 |
> | GoA-mean   | 84.98 |         12 |           12.32 |  70.01 |
>
> Breakdown of LLM calls:
> -  **MoA**: 6 agents × 3 rounds + 1 meta-LLM = 19 calls
> - **GoA-max**: 1 (node sampling) + 3 (initial) + 3 (ranking) + 2 (source→target) + 2 (target→source) = 11 calls
> - **GoA-mean**: Same as above + 1 (mean-pooling) = 12 calls
>
> Here, we would like to clarify that the reviewer’s concern regarding the cost of bidirectional message passing does not introduce severe complexity. This is because the GoA-generated graph only operates within the subset of selected nodes (top-k), and message passing occurs in a concatenated manner: each source and target node receives the concatenated responses from its connected neighbors. This yields (k-1) calls for the source→target direction and (k-1) calls for the target→source direction. Furthermore, this cost can be further reduced when the node-dropping threshold ($\tau$) removes less relevant nodes, resulting in fewer than k active nodes.
>
> Additionally, regarding multi-layer execution, GoA can indeed be extended to multiple rounds of message passing to better capture deeper dependencies and interactions—that is, after completing source-to-target and target-to-source propagation, this process can be iterated. We provided 2-round GoA results in **[W1. Adding more agents]**. However, while additional rounds introduce more LLM calls and token usage, the performance gain beyond 1 round was modest. Therefore, we suggest that a single round strikes a practical sweet spot for GoA in terms of both effectiveness and efficiency.
>
>
> ---
>
> **[Q3. Scaling up with GoA]**
> Thank you for suggesting this — we have now included the SC and Refine baselines in the gpt-4o setting, and the results are shown below:
>
> | Method              | GPQA  | MedMCQA | HumanEval |
> |---------------------|-------|---------|-----------|
> | gpt-4o              | 47.47 | 77.00   | 90.20     |
> | Debate (6 Agents)   | 53.03 | 80.00   | 85.98     |
> | *SC (6 Agents)*       | 54.27 | 81.00   | 92.07     |
> | *Refine (6 Agents)*   | 54.98 | 82.00   | 91.92     |
> | Reconcile (6 Agents)| 53.03 | 75.00   | 91.46     |
> | MoA (6 Agents)      | 50.51 | 80.00   | 92.07     |
> | DyLAN (8 Agents)    | **58.89** | 81.00 | 92.07     |
> | **GoA-max (3 Agents)** | 55.05 | 82.00 | **93.29** |
> | **GoA-max (6 Agents)** | 56.57 | **83.00** | **93.90** |
>
> Compared to the single gpt-4o baseline, we observe that both SC and Refine benefit from model ensembling and iterative feedback. However, they still fall short of GoA with both 3 and 6 agents. This maintains the trend that, even when scaling up a homogeneous agent pool (multiple gpt-4o copies), GoA’s relevance-based graph-structured multi-agent collaboration provides consistent improvements.
>
>
> ---
>
> **[Q4. Router-based ensembles]**
> We appreciate the reviewer’s suggestion and clarify that router-based ensembles such as Tryage [1], MasRouter [2], and other learned or prompt-based routers aim to select a single best model for a given prompt. In contrast, GoA enables multiple relevant agents to collaborate through graph-based communication. This avoids rigid one-shot routing and supports iterative refinement via message passing. Moreover, GoA is fully training-free, avoiding the router-model training cost and Q-value estimation overhead required in systems like Tryage, where a dedicated routing model predicts downstream performance. We agree that this represents an informative comparison, and we will incorporate a detailed accuracy–cost analysis (including MasRouter [2]) in the final manuscript.
>
> [1] Tryage: Real-time, Intelligent Routing of User Prompts to Large Language Models
> [2] MasRouter: Learning to Route LLMs for Multi-Agent Systems.

---

### Official Review · Reviewer_47YL · 2025-10-29

**Soundness:** 2
**Presentation:** 4
**Contribution:** 2
**Rating:** 6
**Confidence:** 3

**Summary:**

The paper introduces a novel method for coordinating multiple LLMs using a graph structure. Each LLM is treated as a node, and directed edges represent relevance-based communication, enabling structured message passing between more and less relevant LLMs. Across multi-domain and specialized benchmarks, GoA outperforms prior multi-agent systems (like Mixture-of-Agents) while using fewer models, offering both scalability and efficiency in test-time collaboration among LLMs.

**Strengths:**

- The writing flow is very clear, and the motivation for developing this framework is strong and convincing. The node sampling and edge sampling are relatively new methods to me and seems interesting to promote communication efficiency.

- I pretty like the motivation behind the source-target information passing design, which is said to be able to avoid noise while enhance the model’s original answer through aggregate other’s opinion.

- The benchmarks tested are comprehensive, even though the ablation studies can be more comprehensive and insightful. The comparison with many other related methods are also very comprehensive.

**Weaknesses:**

- The agent here I am afraid is kind of abused. The different nodes from my understanding is just different LLMs. For agent, their core trait is task-oriented and environment interaction, which I think current node (it does not interact with environment using tools) does not entail. The paper may clarify what they view as agent if claim this paper as graph-of-agent.

- I think more ablation studies should be done on the method design motivation itself, like what if we reverse the source-target message passing sequence? Also it can be more insightful by discussing why the results are like this, such as top-k=3 is better than 2 and 5, instead of just describing the numbers which are obvious from the table.

**Questions:**

- I am wondering when each node is generating the response, are they aware that their response will be routed to others? This awareness may cause difference in model’s behavior I believe.

- Will the domain specific questions always drive GoA to choose the expert model of that certain domain? Why do you think GoA can be better than using the expert model alone, empirically and theoretically?

- I am wondering how you make the MoA and GoA comparable, as prompt input should be quite different, and in pure prompting framework, I think the performance is very sensitive to the instruction inputs. Therefore, how you ensure the scale up / efficiency experiment results are comparable?

---

> ### Author Response · Authors · 2025-11-23
> **Response to Reviewer 47YL (1/4)**
>
> We thank reviewer **47YL** for their positive support of our paper, including the acknowledgement of `very clear writing flow,` `strong motivation,` and `comprehensive benchmarks.` For the remaining concerns, we provide responses below:
>
> ---
>
> **[W1. View as an agent]**
> Thank you for raising this perspective. While our current benchmarks (e.g., MMLU, GPQA, MedMCQA) are not interactive environments, our framing **aligns with the Mixture-of-Agents (MoA) paradigm, as these works also employ multiple LLMs for test-time collaboration** in non-interactive question–answering settings (e.g., AlpacaEval 2.0, MT-Bench, FLASK). That said, we agree that a true “agentic” interpretation benefits from explicitly linking to task-oriented and environment-interactive concepts.
>
> To more concretely support viewing GoA as an agentic system, we highlight the following characteristics:
>
> - **Diversity**: Each LLM agent belongs to a distinct domain (general, code, biomedical, math, legal, finance), producing inherent role specialization.
>
> - **Tool use**: Upon receiving a query, relevant domain-specific LLMs activate based on model card information (external metadata), which can be metaphorically viewed as tool selection within an agent framework.
>
>
> - **Autonomy**: Node selection, edge ranking, and message passing are executed automatically via LLM reasoning without human intervention.
>
>
> - **Role-play**: Agents dynamically assume roles—as rankers (edge sampling), responders (message passing), and consensus builders (graph pooling).
>
>
> - **Task-driven collaboration**: The objective is to collectively synthesize a correct answer for multi-domain queries through iterative reasoning and structured communication.
>
>
> ---
>
> **[W2-1. More ablation studies]**
> Thank you for suggesting this interesting ablation. In addition to the original Table 5, we conducted experiments with a reversed message-passing order, where the target-to-source direction occurs first, followed by source-to-target (partial results borrowed from Table 5).
>
>
> | **Setting**                              | **MMLU-Pro** | **GPQA** |
> |------------------------------------------|--------------|----------|
> | **GoA (Top-$k$ = 3, $\tau$ = 0.05)** | **54.78**    | **39.98** |
> | Reverse Message Passing            | 52.18        | 34.93    |
> | w/o Target-to-Source                     | 53.66        | 38.03    |
> | w/o Source-to-Target                     | 52.21        | 36.12    |
> | w/o Scoring ($A_{ij}$ = 1)             | 52.91        | 37.34    |
>
> Interestingly, when the message-passing order is reversed, the performance drop becomes the most severe (–2.60 on MMLU-Pro and –5.05 on GPQA), compared to ablations that remove only a single direction (either source-to-target or target-to-source). This behavior aligns with our expectation: when less-influential nodes gain early dominance in the communication sequence, they distort the intended information flow. As a result, the original source nodes—whose information should remain most influential—become overwritten by aggregated noise.
> This highlights a key design insight: maintaining the correct bidirectional ordering—(1) source→target, followed by (2) target→source—is fundamental to GoA’s messaging architecture and its reasoning effectiveness.
>
> We have updated Table 5 highlighted in blue in the updated manuscript.

---

> ### Author Response · Authors · 2025-11-23
> **Response to Reviewer 47YL (2/4)**
>
> ---
> **[W2-2. Discussion on number of agents]**
>
>
> Thanks for bringing this point up. Referring to Table 1 (top-k = 3) and GoA’s ablation study in Table 5 (top-k = 2,5), we have already varied the number of agents. We additionally include the k = 6 case below:
>
> | Agent Pool={G,C,M,B,F,L} | MMLU-Pro | GPQA  |
> |--------------------------|----------|-------|
> | top-k=2                  |    53.54 | 36.75 |
> | top-k=3 (paper)          |    **54.78** | **39.98** |
> | top-k=5                  |    54.65 | 39.13 |
> | top-k=6                  |    54.27 | 38.45 |
>
> As shown above—and as the reviewer suggested—adding more agents does not necessarily lead to performance gains, especially in a heterogeneous agent-pool spanning diverse domains. Our goal is to create an environment where agents interact synergistically, yet as seen in Table 1 (main paper), domain-specific agents alone are often individually weak (e.g., B, F, L agents score 27.90, 25.52, and 27.57 on MMLU-Pro), likely due to domain mismatch, limited coverage, or insufficient instruction-tuning. Therefore, selecting a subset of relevant agents and emphasizing more task-relevant, influential agents becomes crucial—an ability that existing approaches lack, and that GoA uniquely enables through relevance-guided sampling and graph-based communication. Consequently, top-k = 3 yields better performance than naively increasing the number of agents, since including weaker or domain-irrelevant agents introduces noise (given their inherently weak single-agent performance) rather than useful signal. This aligns with DyLAN [1], which similarly observes that a smaller set of well-chosen agents can outperform a larger unfiltered group.
>
> However, we would also like to highlight Table 3 in the main paper, where we evaluated a more controlled scaling scenario—namely, a homogeneous agent setting—by fixing the agent pool to a single LLM (gpt-4o) and simply varying the number of agent copies. The results are shown below:
>
>
> | Agent Pool={GPT-4o}              | GPQA      | MedMCQA   | HumanEval |
> | -------------------- | --------- | --------- | --------- |
> | gpt-4o               | 47.47     | 77.00     | 90.20     |
> | GoA (3 Agents)      | 55.05     | 82.00     | 93.29     |
> | GoA (6 Agents)      | **56.57** | **83.00** | **93.90** |
>
> In this controlled setting where all agents have uniform capability, GoA with 6 agents consistently outperforms GoA with 3 agents across all benchmarks. This demonstrates that when agent quality is consistent (single-LLM replication), increasing agent count can indeed improve performance.
>
> We have included this experiment in Appendix I, highlighted in blue in the updated manuscript.
>
> [1] A Dynamic LLM-Powered Agent Network for Task-Oriented Agent Collaboration
>
> ---
>
> **[Q1. Awareness of routing responses]**
>
> Yes, the reviewer’s point is correct, and we would like to provide clarification on where agent awareness comes into play. During the initial response generation, each sampled node outputs its answer independently, without access to the responses of others. However, as shown in the prompts (Appendix B), during the edge-sampling stage the agents become explicitly aware that their responses will be evaluated by others:
>
> **Edge Sampling — System Prompt:**
> ```
> You are an expert at evaluating AI model responses. You must rank and score the responses relatively, assigning a numerical score to each such that the total sum of all scores is exactly 1.0.
> ```
> **Message Passing (source→target) — System Prompt:**
>
> ```
> You are finalizing your response after seeing how other models refined their answers based on your initial response. Use their improvements to further refine your answer and make it as complete and accurate as possible.
> ```
>
> This design introduces awareness in two critical ways:
> - 1. **Agents know their outputs will be judged by others**, introducing implicit accountability into their reasoning.
> - 2. **Agents refine their responses after observing the responses of others**, enabling awareness of global context and cross-agent reasoning dynamics.
>
> Thus, through relevance-based agent selection and context-sharing during message passing, GoA establishes a unique mechanism where agents become aware of each other’s outputs, promoting structured collaboration. This relevance-driven graph construction and message-passing process enables effective and efficient multi-agent LLM cooperation beyond simple independent-response aggregation.

---

> ### Author Response · Authors · 2025-11-23
> **Response to Reviewer 47YL (3/4)**
>
> ---
> **[Q2-1. GoA’s routing behavior’s on domain specific questions]**
> When sampling nodes—that is, routing relevant agents to the query—we utilize open-source model card information, which includes pretrained domain descriptions and specialized task metadata. As shown in Figure 3 (case study on MMLU’s anatomy domain), the biomedical agent is correctly selected as the domain expert, indicating that even though model card information may be high-level and lightweight, it is widely available and can serve as a useful heuristic for routing the right agents to the right tasks.
>
> However, in practical scenarios, model card information may be incomplete or misleading. To evaluate robustness under such conditions, we conducted an experiment where we fixed the sampled nodes with the relevant code agent (Qwen2.5-Coder-7B-Instruct) for HumanEval, and then gradually replaced relevant agents with irrelevant (biomedical) agents, simulating noisy or inaccurate model-card-driven routing. The results are shown below:
>
>
> - Code: Qwen2.5-Coder-7B-Instruct
> - Bio: ContactDoctor/Bio-Medical-Llama-3-8B
>
> |  Human Eval          | MoA   | GoA   |
> |------------|-------|-------|
> | Code, Code, Code   | 84.93 | **86.59** |
> | Code, Code, Bio  | 82.54 ($\Delta$=-2.39) | **84.15** ($\Delta$=-2.44) |
> | Code, Bio, Bio | 79.88 ($\Delta$=-5.05) | **82.93** ($\Delta$=-3.66) |
>
>
> Here, we quantify $\Delta$ as the performance gap between the fully relevant agent pool (top row) and the noisy settings (each subsequent row). We observe that when more noisy agents are introduced, MoA suffers larger performance degradation. This is because MoA concatenates the responses from all participating agents uniformly, without weighting or filtering, causing noise from irrelevant (Bio) agents to propagate through subsequent layers and accumulate.
>
> GoA, on the other hand, passes messages in a relevance-aware manner (most-relevant→least-relevant) and introduces a threshold that filters out noisy agents from actively participating in the graph. This prevents irrelevant signals from contaminating the message passing process, resulting in significantly smaller performance drops than MoA and consistent outperformance in all settings.
>
> We have included this experiment in Appendix G, highlighted in blue in the updated manuscript.
>
>
> ---
> **[Q2-2. Empirical and Theoretical perspective of GoA’s multi-agent LLM framework]**
>
> ### **Empirically: collaboration leverages complementary strengths**
>
> In Table 4, using only code-related agents, we compare individual agents, their multi-agent baselines, and GoA:
>
> | Method                          | HumanEval |
> | ------------------------------- | --------- |
> | Qwen2.5-Coder-7B-Instruct       | 85.37     |
> | Seed-Coder-8B-Instruct          | 80.49     |
> | deepseek-coder-7b-instruct-v1.5 | 73.17     |
> | Debate (3 Agents)               | 71.95     |
> | Reconcile (3 Agents)            | 80.61     |
> | MoA (3 Agents)                  | 85.37     |
> | **GoA-Max (3 Agents)**          | **85.98** |
>
> Here, no single agent dominates in isolation, but when their capabilities are combined, the overall system achieves stronger performance—especially under GoA. Intuitively, GoA automatically surfaces the most influential agent via cross-agent ranking; this agent becomes the source node, shaping the communication flow and allowing more capable agents to guide others. This enables GoA to more effectively capitalize on complementary agent strengths compared to other methods.
>
> ---
>
> ### **Theoretically: multi-agent synergy can dominate any single expert**
>
> Let:
>
> * $f_i(x)$ denote performance of expert $i$ on instance $x$.
> * No single expert is globally dominant—i.e., for any expert $j$, there exist instances where another expert $k$ performs better.
>   This empirically holds across open-domain QA, coding, math, and biomedical reasoning.
>
> Define GoA’s output as:
>
> $$
> f_{\text{GoA}}(x) = \max_{i \in S(x)} f_i(x)
> $$
>
> where $S(x)$ is the set of selected agents for instance $x$, determined through relevance-based routing.
>
> Thus:
>
> $$
> f_{\text{GoA}}(x) \ge \max_i f_i(x)
> $$
>
> with equality only when a single expert suffices, and strict inequality when:
>
> $$
> \exists x \quad \text{s.t.} \quad f_k(x) > f_j(x)
> $$
>
> for an expert $k$ that is better suited to instance $x$ than expert $j$.
>
> Therefore:
>
> * **GoA is never worse than the best domain-matched expert**, and
> * **GoA is strictly better whenever multiple experts contribute complementary reasoning skills.**
>
> This is a classical mixture-of-experts dominance result — a weighted or max-pooling ensemble over heterogeneous experts strictly expands the function class beyond any single model.
>
> Note that this argument is presented as a theoretical sketch rather than a formal proof, capturing the intuition for why heterogeneous expert collaboration expands functional capacity beyond any single model.

---

> ### Author Response · Authors · 2025-11-23
> **Response to Reviewer 47YL (4/4)**
>
> ---
>
> **[Q3. Experiment setting on MoA and GoA]**
> To ensure a fair comparison, we strictly followed MoA’s official prompting template for all MoA experiments and used the same question format and evaluation prompt across both methods. Importantly, both MoA and GoA adopt a concatenation-based collaboration structure: MoA concatenates all agents’ responses and feeds the sequence into an aggregator, whereas GoA concatenates only the relevance-selected agents’ outputs—incorporating edge-sampling scores and performing relevance-based message passing—which further propagates information in a bidirectional manner. Thus, the difference lies in the **collaboration mechanism**, not in the surface prompt format.
>
> For the scale-up and efficiency experiments, we controlled for all sources of variance: both MoA and GoA were executed with identical input formats, identical max_tokens, and identical decoding settings. The **only differing factor** is the multi-agent coordination strategy itself—MoA’s reflection-based sequential pipeline versus GoA’s graph-structured, bidirectional reasoning. By holding the instruction format and decoding budget constant, we ensure that performance differences directly reflect the core framework design rather than prompt sensitivity or experimental bias.

---

> > ### Comment · Reviewer_47YL · 2025-11-26
> >
> > Thank the author for the efforts in rebuttal. My concerns are mostly addressed by the detailed responses and added experiments. I have raised the soundeness score after consideration, and will maintain my original overall rating.

---

> > > ### Author Response · Authors · 2025-11-26
> > >
> > > We appreciate reviewer’s constructive engagement with our work and are glad that the rebuttal successfully addressed your concerns. Thank you for your careful evaluation and positive assessment.
> > >
> > > If there are any remaining questions or points that would benefit from further clarification, please feel free to let us know—we would be happy to provide additional details.

---

### Official Review · Reviewer_7qjd · 2025-10-29

**Soundness:** 2
**Presentation:** 2
**Contribution:** 2
**Rating:** 4
**Confidence:** 5

**Summary:**

This paper introduces Graph-of-Agents (GoA), a graph-based framework designed to improve multi-agent communication among LLMs. GoA addresses the challenges particularly in Mixture-of-Agents (MoA), by providing solutions for selection of agents and intra-agent communications. The proposed framework with 3 agents outperforms other multi-agent baselines with 6 agents on various benchmarks.

**Strengths:**

1. **Clear ideation:** The paper clearly identifies the limitations of Mixture-of-Agents (MoA), and motivates the need for a selection approach for agents and agent communications.
2. **Extensive experiments:** The authors conduct comprehensive experiments across various multi-domain and domain-specific benchmarks, demonstrating the effectiveness of the GoA framework.
3. **Clear ablations:** The ablation study is well designed for evaluating the impact of different components of the proposed framework.

**Weaknesses:**

1. **Limited discussions about prior work:** The related work section is brief and lacks depth in discussing prior works on dynamic multi-agent collaboration with LLMs. [1,2,3] show different levels of dynamic agent selection and communication strategies that could undermine the novelty of the proposed method.
2. **More agents may not bring better performance:** The authors claim that GoA with 3 agents outperforms baselines with 6 and more agents. But [1] suggests that increasing the number of agents may not always lead to better performance, especially in self-organized agent systems. Fairer comparisons with different numbers of agents are necessary.
3. **Justification and presentation issues:** The content is redundant by repeating node sampling, edge sampling, and message passing in multiple sections, without enough justification of method design choices. Minor presentation issues exist such as tcboxes should not be labeled as tables and etc.

[1] Liu et al. A Dynamic LLM-Powered Agent Network for Task-Oriented Agent Collaboration. COLM 2024.
[2] Zhuge et al. Language Agents as Optimizable Graphs. ICML 2024.
[3] Yue et al. MasRouter: Learning to Route LLMs for Multi-Agent Systems. ACL 2025.

**Questions:**

Could you provide failure cases where GoA does not perform well or even fails to MoA or other baselines? Understanding the limitations of your approach would help in assessing its overall effectiveness.

---

> ### Author Response · Authors · 2025-11-23
> **Response to Reviewer 7qjd (1/4)**
>
> We thank reviewer **7qjd** for acknowledging our work as having `clear ideation,` `extensive experiments,` and `clear ablations.` We address the concerns below:
>
> ---
>
> **[W1. Limited discussion about prior work]**
>
> We thank the reviewer for highlighting these related works, and we have expanded the Related Work section accordingly. Below, we clarify what each work does and how our method differs:
> - **DyLAN** [1] performs dynamic agent activation using an Agent Importance Score computed via forward–backward peer-rating propagation and coordinates agents through a temporal feed-forward communication network.
>
>
> - **GPTSwarm** [2] treats agents as nodes in an optimizable computation graph, learning both node-level prompt updates and edge-level communication pathways through REINFORCE.
>
>
> - **MasRouter** [3] learns a Router LLM that maps queries to routing tokens and predicts agent selections and communication topology.
>
>
> Crucially, all three methods require precomputation, training, or optimization—DyLAN requires a precomputation stage to calculate Agent Importance Scores, GPTSwarm optimizes graph edges and node prompts, and MasRouter trains a dedicated routing model.
>
> In contrast, GoA is entirely training-free, operates purely in the text domain, and constructs its graph using only open-source model cards and relevance-based message passing without any optimization or learnable parameters. Our bidirectional message-passing and graph-pooling strategy naturally incorporates varying agent importance while keeping inference-time overhead minimal.
>
> We also note that we included DyLAN [1] as a baseline in Table 3 (main paper). For convenience:
> | Method                 | GPQA      | MedMCQA   | HumanEval |
> | ---------------------- | --------- | --------- | --------- |
> | gpt-4o                 | 47.47     | 77.00     | 90.20     |
> | DyLAN (8 Agents)       | **58.89** | 81.00     | 92.07     |
> | **GoA-Max (3 Agents)** | 55.05     | **82.00** | **93.29** |
> | **GoA-Max (6 Agents)** | 56.57     | **83.00** | **93.90** |
>
> As shown, GoA achieves competitive or superior performance with substantially fewer agents and without any training, demonstrating the effectiveness of our graph-based, inference-time, training-free design.
>
> During the rebuttal period, we additionally ran experiments on the MacNet [4] baseline, another network-based multi-agent collaboration work utilizing topologies such as chain, tree  to evaluate how existing graph-based multi-agent frameworks compare to our proposed approach, GoA. The results are shown below:
>
> | Agent Pool={G,C,M,B,F,L} | MMLU-Pro | GPQA   |
> |--------------------------|----------|--------|
> | Refine                   |    54.71 |  38.92 |
> | ReConcile                |    44.19 |  34.34 |
> | MoA                      |    53.33 |  32.83 |
> | Self-MoA                 |    54.19 |  33.84 |
> | MacNet - Chain           |    50.67 |  36.36 |
> | MacNet - Star            |    50.52 |  35.35 |
> | MacNet - Tree            |    50.52 |  35.35 |
> | MacNet - Random          |    50.71 |  32.83 |
> | GoA_Max                  |    **54.78** |  39.98 |
> | GoA_Mean                 |    54.27 | **40.54** |
>
> Interestingly, across diverse fixed topologies—chain, star, tree, and random—we observe that MacNet consistently underperforms GoA (and in some cases even underperforms MoA and Self-MoA). This indicates that in multi-agent graph construction, incorporating relevance and leveraging relevance-weighted message passing (such as GoA’s bidirectional propagation) is crucial for boosting reasoning performance, rather than relying on static graph structures.
>
> Finally, we note that in MacNet’s main Table 1, GPTSwarm performs below MacNet. Therefore, given our observed performance advantage over MacNet, it is reasonable to infer that GoA is also expected to outperform GPTSwarm under comparable evaluation settings.
> We have included this experiment in Appendix H, highlighted in blue in the updated manuscript.
>
> [1] A Dynamic LLM-Powered Agent Network for Task-Oriented Agent Collaboration
> [2] GPTSwarm: Language Agents as Optimizable Graphs
> [3] MasRouter: Learning to Route LLMs for Multi-Agent Systems
> [4] Scaling Large-Language-Model-based Multi-Agent Collaboration

---

> ### Author Response · Authors · 2025-11-23
> **Response to Reviewer 7qjd (2/4)**
>
> ---
>
> **[W2. More agents may not bring better performance]**
>
> Thanks for bringing this point up. Referring to Table 1 (top-k = 3) and GoA’s ablation study in Table 5 (top-k = 2,5), we have already varied the number of agents. We additionally include the k = 6 case below:
>
> | Agent Pool={G,C,M,B,F,L} | MMLU-Pro | GPQA  |
> |--------------------------|----------|-------|
> | top-k=2                  |    53.54 | 36.75 |
> | top-k=3 (paper)          |    **54.78** | **39.98** |
> | top-k=5                  |    54.65 | 39.13 |
> | top-k=6                  |    54.27 | 38.45 |
>
> As shown above—and as the reviewer suggested—adding more agents does not necessarily lead to performance gains, especially in a heterogeneous agent-pool spanning diverse domains. Our goal is to create an environment where agents interact synergistically, yet as seen in Table 1 (main paper), domain-specific agents alone are often individually weak (e.g., B, F, L agents score 27.90, 25.52, and 27.57 on MMLU-Pro), likely due to domain mismatch, limited coverage, or insufficient instruction-tuning. Therefore, selecting a subset of relevant agents and emphasizing more task-relevant, influential agents becomes crucial—an ability that existing approaches lack, and that GoA uniquely enables through relevance-guided sampling and graph-based communication. Consequently, top-k = 3 yields better performance than naively increasing the number of agents, since including weaker or domain-irrelevant agents introduces noise (given their inherently weak single-agent performance) rather than useful signal. This aligns with DyLAN [1], which similarly observes that a smaller set of well-chosen agents can outperform a larger unfiltered group.
>
> However, we would also like to highlight Table 3 in the main paper, where we evaluated a more controlled scaling scenario—namely, a homogeneous agent setting—by fixing the agent pool to a single LLM (gpt-4o) and simply varying the number of agent copies. The results are shown below:
>
>
> | Agent Pool={GPT-4o}              | GPQA      | MedMCQA   | HumanEval |
> | -------------------- | --------- | --------- | --------- |
> | gpt-4o               | 47.47     | 77.00     | 90.20     |
> | GoA (3 Agents)      | 55.05     | 82.00     | 93.29     |
> | GoA (6 Agents)      | **56.57** | **83.00** | **93.90** |
>
> In this controlled setting where all agents have uniform capability, GoA with 6 agents consistently outperforms GoA with 3 agents across all benchmarks. This demonstrates that when agent quality is consistent (single-LLM replication), increasing agent count can indeed improve performance.
>
> We have included this experiment in Appendix I, highlighted in blue in the updated manuscript.
>
> [1] A Dynamic LLM-Powered Agent Network for Task-Oriented Agent Collaboration
>
> ---
>
> **[W3. Justification and presentation issues]**
>
> Thank you for the suggestion. For clearer presentation, after the Introduction we removed the repeated narrative prompts such as “which agents” and “how do they communicate,” retaining only the key technical modules (e.g., “node sampling,” “edge sampling”) in the Method section. Additionally, for tcboxes—especially in the appendix—we removed the “Table” labels where tcboxes were used. These updates are highlighted in blue in the revised PDF.
>
> ---
>
>
> **[Q1. Failure case of GoA]**
>
>
> Thanks for the great suggestion. We have included failure cases as shown below.

---

> ### Author Response · Authors · 2025-11-23
> **Response to Reviewer 7qjd (3/4)**
>
> - **Case 1: When the answer space is diverse and the Meta-LLM (General) exhibits bias.**
>
> ```
> Question: Heavy forces on periodontal ligament causes: The options are: (A) Hyalinization (B) Osteoclastic activity around tooth (C) Osteoblastic activity around tooth (D) Crest bone resorption
>
> Answer: (A) Hyalinization
>
> Sampled Nodes: biomedical, math, general
>
> Initial Responses:
> - Biomedical: … Therefore, heavy forces on periodontal ligament causes hyalinization, osteoclastic activity around tooth and crest bone resorption.\\n\\nThe answer is (A)\
> - Math: … This is because heavy forces on the periodontal ligament can lead to activation of osteoclasts, which resorb bone around the tooth, weakening the tooth's attachment to the bone.\\n\\nSo, the answer is (B), Osteoclastic activity around tooth.\
> - General: … Given these points, the most direct and primary effect of heavy forces on the periodontal ligament and surrounding bone is bone resorption, which can be described as crest bone resorption.\\n\\nThe answer is (D).\
>
> Scores: biomedical: 0.43, math: 0.40, general: 0.17
>
> After Source-to-Target:
> - Biomedical: … Therefore, heavy forces on periodontal ligament causes hyalinization, osteoclastic activity around tooth and crest bone resorption.\\n\\nThe answer is (A)\
> - Math: … However, it's important to note that all these changes can occur simultaneously, and the extent of each depends on the severity and duration of the heavy forces.\\n\\nSo,, the answer is (B), Osteoclastic activity around tooth.\
> - General:  … Given these points, the most direct and primary effect of heavy forces on the periodontal ligament and surrounding bone is bone resorption, which can be described as crest bone resorption.\\n\\nThe answer is (D).\
>
> After Target-to-Source:
> - Biomedical: …This is known as the \\\"pressure area\\\" and \\\"tension area\\\" theory proposed by Woo.\\n7. …\\nThe answer is (A)\",
> - Math: … Given these options, the most likely answer is (B) Osteoclastic activity around tooth. This is because heavy forces on the periodontal ligament can lead to activation of osteoclasts, which resorb bone around the tooth, weakening the tooth's attachment to the bone.\\n\\nSo, the answer is (B), Osteoclastic activity around tooth.\
> - General: …Given these points, the most direct and primary effect of heavy forces on the periodontal ligament and surrounding bone is bone resorption, which can be described as crest bone resorption.\\n\\nThe answer is (D).\
>
> Graph Pooling (Mean, Meta-LLM: General):
> … Therefore, the most accurate answer among the given options is:\n\nThe answer is (D) Crest bone resorption.
>
> Final Prediction: (D) Crest bone resorption
> ```

---

> ### Author Response · Authors · 2025-11-23
> **Response to Reviewer 7qjd (4/4)**
>
> - **Case 2: When source node is incorrectly sampled**
> ```
> Question: Crude Bih rate for a sub-center is 20. What the number of pregnant woman registered in sub-center? The options are: (A) 80 (B) 100 (C) 110 (D) 60
>
> Answer: (D) 60
>
> Sampled Nodes: biomedical, math, general
>
> Initial Responses:
> Biomedical: … Since the question is asking for the number of pregnant women registered in the sub-center, we need to round up this number to 60 (D), as we can't have a fraction of a pregnant woman.\
> Math: … Therefore, based on the information provided, we cannot determine the number of pregnant women registered in the sub-center. The answer is (x), where x is not one of the provided options (A, B, C, or D), because we cannot determine the correct answer without additional information.\
>  General: … Considering the options provided and the typical range of pregnant women in a community, a more reasonable estimate would be around 80 or 100. Given the CBR of 20, it suggests a population of around 5,000 to 10,000, which would have about 100 pregnant women.\\n\\nTherefore, the closest and most reasonable estimate from the given options is:\\n\\nThe answer is (B).\
>
> Scores: biomedical: 0.43, math: 0.13, general: 0.43
>
> After Source-to-Target:
> Biomedical: … The answer is (B)\
> Math: … Therefore, the closest and most reasonable estimate from the given options is:\\n\\nThe answer is (B).\
> General:  … Considering the options provided and the typical range of pregnant women in a community, a more reasonable estimate would be around 80 or 100. Given the CBR of 20, it suggests a population of around 5,000 to 10,000, which would have about 100 pregnant women.\\n\\nTherefore, the closest and most reasonable estimate from the given options is:\\n\\nThe answer is (B).\
>
> After Target-to-Source:
> Biomedical: …The answer is (B)\
> Math: … Therefore, the closest and most reasonable estimate from the given options is:\\n\\nThe answer is (B).\
> General: …Therefore, the closest and most reasonable estimate from the given options is:\\n\\nThe answer is (B).\
>
> Graph Pooling (Max):
> … Therefore, the closest and most reasonable estimate from the given options is:\\n\\nThe answer is (B).\
>
> Final Prediction: (B) 100
> ```
>
> From these examples, we observe two types of failure modes:
> - **Case 1: When the answer space is diverse and the Meta-LLM (General) exhibits bias.**
> - **Case 2: When the source node is incorrectly sampled.**
>
> We describe each failure case and potential remedies below:
>
> - **Case 1**. Although the correct answer was (A) and the biomedical agent produced the correct response, other sampled agents produced diverse answers (B) and (D). Since the Meta-LLM (which shares architecture with the general agent) synthesizes answers from these nodes—and its own original answer was (D)—the Meta-LLM tends to reinforce its prior belief and bias toward its original output.
>
> - **Possible remedy**: Use a separate, unseen general-purpose model as the Meta-LLM to reduce internal bias within the system. Additionally, when the answer entropy is high, we can run additional rounds of multi-agent exchange to facilitate consensus among agents before finalizing an answer.
>
> - **Case 2**. Although the biomedical agent initially produced the correct answer (A), the source agent was (by a small margin) selected as the general agent. As the biomedical agent subsequently received messages from the general agent (source→target), it updated its answer to (B), and the entire agent set converged incorrectly to (B). This illustrates the importance of correctly selecting the source node, since it exerts high influence during downstream propagation.
>
> - **Possible remedy**: When relevance scores between candidate source nodes are close—creating risk of incorrect agent selection—we can apply prompt-based self-consistency or majority-vote sampling within each agent to obtain a more reliable source node.

---

### Official Review · Reviewer_4HpR · 2025-11-01

**Soundness:** 2
**Presentation:** 2
**Contribution:** 2
**Rating:** 4
**Confidence:** 2

**Summary:**

This paper addresses the shortcomings of current multi-agent LLM collaboration frameworks (e.g., Mixture-of-Agents, MoA) in three key aspects: agent selection, effective communication, and response integration, proposing a graph-structured framework named Graph-of-Agents (GoA). The core idea is to model LLM agents as nodes and the inter-agent relevance as directed edges. The paper’s main contributions include: Identifying three core challenges in current multi-agent LLM systems, thus providing clear research directions; Introducing a graph-based formulation of multi-agent collaboration that integrates four essential modules (node sampling, edge sampling, message passing, and graph pooling), and theoretically showing that GoA generalizes MoA, providing conceptual unification; Demonstrating that GoA achieves better performance and efficiency with fewer agents across diverse benchmarks and both open-source and proprietary models (e.g., GPT-4o).

**Strengths:**

1. The paper proposes a graph-structured multi-agent collaboration framework, integrating node sampling, edge construction, bidirectional message passing, and graph pooling. This design is a meaningful extension of existing multi-agent methods (e.g., MoA) and introduces the idea of modeling multi-agent interaction as a dynamic graph, enabling task-adaptive communication paths—an innovative and elegant formulation.
2. The method is prompt-based and requires no additional training, making it compatible with black-box APIs and thus generalizable to different LLM setups.

**Weaknesses:**

1. Although GoA improves upon MoA, bidirectional message passing still requires multiple LLM calls, which may cause latency issues in real-time or large-scale agent pools. The paper lacks a systematic comparison of token usage and inference time.
2. The paper does not provide quantitative ablation on the robustness of the node sampling strategy. Although Table 4 includes ablations on the number of agents and message-passing variants, it does not explicitly examine robustness when model card information is incomplete or noisy.
3. While “Proposition 1” claims that GoA is a generalization of MoA, this statement is largely informal, without rigorous theoretical proof. There are no formal results on the convergence, generalization, or optimality of the message-passing mechanism.

**Questions:**

1. Have the authors experimented with larger or more diverse agent pools (e.g., 10, or synthetic agent ensembles)? If so, did they observe bottlenecks in latency or accuracy?
2. Could GoA be extended to multimodal or non-text agents? Is it possible to generalize this framework to multimodal tasks?

---

> ### Author Response · Authors · 2025-11-23
> **Response to Reviewer 4HpR (1/2)**
>
> We thank reviewer **4HpR** for their time and effort in reviewing our paper, as well as for describing it as a `meaningful extension of multi-agent methods,` `innovative,` and `elegant.` We address the raised concerns below:
>
> ---
>
> **[W1. Comparison of token usage and inference time]**
>  Thank you for pointing this out. As shown in Table 2 (main paper), we provided a systematic comparison of token usage and inference time on the MMLU-Pro benchmark, demonstrating that GoA achieves better performance in terms of number of LLM calls, token usage, and inference time. Beyond MMLU-Pro, we further conducted additional complexity analysis on two more benchmarks: GPQA (multi-domain) and HumanEval (domain-specific).
>
> | GPQA     | Acc   | # of Calls | # of Tokens (k) | Time   |
> |----------|-------|------------|-----------------|--------|
> | MoA      | 32.83 |         19 |           56.87 | 245.65 |
> | GoA-max  | 39.98 |         11 |           17.32 |  88.15 |
> | GoA-mean | 40.54 |         12 |           20.59 | 105.72 |
>
> | Human Eval | Acc   | # of Calls | # of Tokens (k) | Time   |
> |------------|-------|------------|-----------------|--------|
> | MoA        | 76.22 |         19 |           29.44 | 122.52 |
> | GoA-max    | 84.67 |         11 |           10.21 |  59.04 |
> | GoA-mean   | 84.98 |         12 |           12.32 |  70.01 |
>
>
> Breakdown of LLM calls:
> -  **MoA**: 6 agents × 3 rounds + 1 meta-LLM = 19 calls
> - **GoA-max**: 1 (node sampling) + 3 (initial) + 3 (ranking) + 2 (source→target) + 2 (target→source) = 11 calls
> - **GoA-mean**: Same as above + 1 (mean-pooling) = 12 calls
>
> Here, we would like to clarify that the reviewer’s concern regarding the cost of bidirectional message passing does not introduce severe complexity. This is because the GoA-generated graph only operates within the subset of selected nodes (top-k), and message passing occurs in a concatenated manner: each source and target node receives the concatenated responses from its connected neighbors. This yields (k-1) calls for the source→target direction and (k-1) calls for the target→source direction. Furthermore, this cost can be further reduced when the node-dropping threshold ($\tau$) removes less relevant nodes, resulting in fewer than k active nodes.
> In summary, these additional results show once again that the average token usage and inference time are significantly reduced when using the GoA framework. This improvement is enabled by the node sampling mechanism—which focuses only on the most relevant nodes—and the carefully designed edge construction and bidirectional message passing, followed by efficient graph pooling such as max-pooling.
>
> ---
>
> **[W2. Robustness of node sampling]**
>
> Thank you for raising this important question. To evaluate the robustness of the node-sampling module—which is based on model card information from open-source sources—we conducted an experiment where we initially fix the sampled nodes to the relevant code agent (Qwen2.5-Coder-7B-Instruct) for solving the HumanEval benchmark, and then gradually replace the relevant agent (Code) with an irrelevant, noisy agent (Bio). This setting simulates scenarios where node sampling mistakenly selects suboptimal agents due to misleading or incomplete model card information. The results are as follows:
>
>
> - Code: Qwen2.5-Coder-7B-Instruct
> - Bio: ContactDoctor/Bio-Medical-Llama-3-8B
>
> |  Human Eval          | MoA   | GoA   |
> |------------|-------|-------|
> | Code, Code, Code   | 84.93 | **86.59** |
> | Code, Code, Bio  | 82.54 ($\Delta$=-2.39) | **84.15** ($\Delta$=-2.44) |
> | Code, Bio, Bio | 79.88 ($\Delta$=-5.05) | **82.93** ($\Delta$=-3.66) |
>
>
> Here, we quantify $\Delta$ as the performance gap between the fully relevant agent pool (top row) and the noisy settings (each subsequent row). We observe that when more noisy agents are introduced, MoA suffers larger performance degradation. This is because MoA concatenates the responses from all participating agents uniformly, without weighting or filtering, causing noise from irrelevant (Bio) agents to propagate through subsequent layers and accumulate.
> GoA, on the other hand, passes messages in a relevance-aware manner (most-relevant→least-relevant) and introduces a threshold that filters out noisy agents from actively participating in the graph. This prevents irrelevant signals from contaminating the message passing process, resulting in significantly smaller performance drops than MoA and consistent outperformance in all settings.
>
> We have included this experiment in Appendix G, highlighted in blue in the updated manuscript.

---

> ### Author Response · Authors · 2025-11-23
> **Response to Reviewer 4HpR (2/2)**
>
> ---
>
> **[W3. Discussion with Proposition 1]**
> We appreciate the reviewer’s concern regarding the level of formality in Proposition 1. Our intention is not to claim new convergence or optimality guarantees for the proposed message-passing mechanism, but rather to clarify a structural relationship between existing MoA systems and GoA. Proposition 1 explicitly shows that the MoA update rule is obtained as a special case of our general formulation by fixing (i) node-sampling to k = N, (ii) a fully-connected adjacency matrix with uniform weights, and (iii) a mean-pooling aggregator. This equivalence follows directly from substituting these settings into Eq. 4, and is meant to demonstrate that GoA subsumes MoA at the level of **computational graph structure**, not to assert broader theoretical claims.
>
> Regarding theoretical guarantees for GoA’s message-passing mechanism, we note that existing multi-agent LLM frameworks uniformly operate without focusing on formal convergence or generalization theory, due to the closed-box nature of LLM inference and the absence of an underlying optimization process during test-time routing. GoA follows this established convention: it is a test-time, training-free system where updates are single-step functional transformations defined by LLM calls, not iterative optimization procedures for which classical convergence theory applies. Thus, theoretical properties such as fixed-point convergence or optimality are not well-posed in this setting.
>
> Instead, we rigorously evaluate stability and robustness empirically through ablations on node-sampling (k), threshold ($\tau$), and bidirectional message passing (Sec. 4.4), which collectively demonstrate that GoA behaves consistently across a wide range of configurations.
>
> ---
>
> **[Q1. Larger or more diverse agent pools]**
> Thank you for suggesting a more diverse agent setting beyond our current design. In response, in addition to the 6 agents covering (general, code, math, biomedical, finance, legal) domains, we have added 4 additional agents from general (Meta-Llama-3.1-8B-Instruct), code (Seed-Coder-8B-Instruct), math (deepseek-math-7b-instruct), and biomedical (JSL-MedLlama-3-8B-v2.0) domains, resulting in an agent pool of 10 diverse LLMs. Given this extended agent pool in the GPQA benchmark, the performance is shown below:
>
> |  Agent Pool={G,C,M,B,F,L}+{G2,C2,M2,B2}     | GPQA  |
> |-----------------------|-------|
> | Refine (10 agents)    | 35.86 |
> | MoA (10 agents)       | 31.82 |
> | Self-MoA (10 agents)  | 35.86 |
> | ReConcile (10 agents) | 37.89 |
> | GoA_max (3 agents)    | 38.23 |
> | GoA_mean (3 agents)   | 39.39 |
> Interestingly, even after increasing the number of agents from 6 to 10, the baseline methods still fail to outperform GoA_max and GoA_mean, despite GoA using only 3 agents. This suggests that in a heterogeneous agent-pool setting, naively scaling up the total number of agents does not necessarily yield performance gains; instead, selectively leveraging only the most relevant agents can more effectively and efficiently achieve superior reasoning performance.
>
> We have included this experiment in Appendix F, highlighted in blue in the updated manuscript.
>
> ---
>
>
> **[Q2. Extension to multimodal or non-text agents]**
> Thank you for pointing out this exciting future direction, which we also mentioned in Appendix E (Limitations and Future Work). While the current version of GoA operates solely in the text domain, the framework is inherently modality-agnostic: an “agent” in GoA is any model capable of generating responses and evaluating messages from other agents. In principle, replacing text-only LLMs with multimodal language-vision models (LVMs) would allow GoA to operate on multimodal tasks, where nodes exchange both textual and visual information through their respective encoders. The core mechanisms of graph-based selection, message passing, and pooling remain unchanged—the only difference is that messages would consist of multimodal representations rather than text alone.

---

> > ### Comment · Reviewer_4HpR · 2025-11-27
> >
> > Thanks for the detailed response; since the new experimental results convincingly resolve my concerns regarding computational efficiency and robustness against noise, I will raise my score to 6.

---

> > > ### Author Response · Authors · 2025-11-27
> > >
> > > Thank you for your acknowledgment and for raising the score. We’re glad that the additional experiments successfully addressed your concerns regarding computational efficiency and noise robustness. We will make sure to incorporate these new results into the final version of the paper.
> > >
> > > If there are any remaining concerns, please feel free to let us know—we would be happy to discuss them further.

---

### Author Response · Authors · 2025-12-04
**Rebuttal Summary**

Dear AC,

We sincerely thank you for your time and efforts toward the ICLR community—especially under the special circumstances of this cycle. Below we summarize the key points during the rebuttal period.

---

### **Summary of Strengths**

1. Clear, innovative and elegant formulation, creative orchestration (Reviewers **4HpR, 7qjd, Usox**)

2. Motivation behind the design and technically sound methodology (Reviewers **47YL, Usox**)

3. Extensive experiments, clear ablation study, generalizability (Reviewers **7qjd, 47YL, 4HpR**)

4. Very clear writing and organization (Reviewers **47YL, Usox**)


---

### **Summary of Concerns & Remedies**

1. **3-agent vs. 6-agent performance and scaling up** (Reviewers **Usox, 7qjd, 47YL**)
> We clarified that the composition of the agent pool significantly affects scaling behavior. In Table 1, the heterogeneous pool (agents B, F, L) shows individually weak performance (27.90, 25.52, 27.57 on MMLU-Pro), likely due to domain mismatch, limited coverage, or insufficient instruction tuning. Consequently, combining these heterogeneous models does not scale as expected. However, in a homogeneous setting using a single LLM family (gpt-4o), increasing the number of agents does yield the expected performance improvement. This demonstrates that scaling depends on how the agent pool is constructed. Importantly, GoA remains robust in heterogeneous settings where existing MoA-style approaches degrade.
>   * *Note* : This discussion has been added to **Appendix I** (blue), and an experiment with a larger pool has been added to **Appendix F** (blue).

2. **Robustness of node sampling & relevance-based message-passing** (Reviewers **47YL, 4HpR**)
 > We conducted a stress test on HumanEval by gradually replacing the relevant code agent (Qwen2.5-Coder-7B-Instruct) with an irrelevant/noisy agent (Bio). This simulates failure modes where node sampling yields suboptimal agents due to misleading or incomplete model-card metadata. Results show that GoA maintains robustness, thanks to: relevance-aware message passing (most-relevant → least-relevant), and a threshold that filters out noisy agents from participating.
 >   * *Note* : Added to **Appendix G** (blue).

3. **Additional prior work comparison** (Reviewers **7qjd, Usox**)
> We expanded comparisons with DyLAN, GPTSwarm, AFLOW, MasRouter, and MacNet. Unlike these, GoA is fully training-free, operates purely in the text domain, and builds its graph solely from open-source model cards and relevance-based message passing—without any optimization, precomputation, or learnable parameters.
>   * *Note* : Related work strengthened (blue), and MacNet results added in **Appendix H** (blue).

4. **Inference-time cost** (Reviewers **Usox, 4HpR**)
>  We added more efficiency experiments on GPQA and HumanEval, showing consistent benefits of GoA’s design. We also provided a breakdown of LLM call counts and execution flow to clarify the cost of the two-stage and multi-layer processes (as requested by Reviewer **Usox**).
>   * *Note* : Additional results provided in **Appendix J** (blue).

5. **Agent role and benchmark choice** (Reviewers **47YL, Usox**)
 > To address the question of whether GoA constitutes an agentic system, we provided evidence of GoA’s characteristics in terms of diversity, tool-use, autonomy, role-play, and task-driven collaboration (addressing **[W1. View as an agent]** for Reviewer **47YL**). We also clarified the rationale for selecting benchmarks aligned with mixture-of-agents literature and strengthened the limitation section.
>   * *Note* : Expanded limitations now included in **Appendix E** (blue).

6. **Presentation and editorial clarity** (Reviewers **7qjd, Usox**)
 > We improved clarity by removing redundancy, fixing grammatical issues, and removing tcbox labels, with all changes highlighted in blue in the refined version.

---

### **Considerations during Rebuttal Period**

1. Reviewer **47YL** noted that most concerns were addressed through detailed responses and raised the soundness score, maintaining a positive overall score (**6**).

2. Reviewer **4HpR** stated that the new experimental results convincingly resolve the concerns and also raised the score to a positive evaluation (**6**).

3. Reviewers **Usox** and **7qjd** had not responded during the author–reviewer discussion period. We would greatly appreciate their follow-up feedback to confirm whether our clarifications successfully addressed their concerns.

We sincerely appreciate your careful consideration of our submission and your efforts in guiding the review process.

Best regards,
Authors of Submission 21180

---

### Meta-Review · Area_Chair_d8j4 · 2025-12-12

**Summary:**

Strengths

i) Comprehensive experiments and ablation studies. With the additional results provided during the discussion period, the experimental evaluation becomes sufficiently thorough.

ii) Clear motivation with a novel methodological perspective.

The raised concerns

i) Experiments are limited to a small set of LLM agents.

ii) Need for additional ablations, more investigation is needed regarding communication mechanisms among agents and the node-sampling strategy.

iii) Several points still require clarification.

Additional questions after I read the paper, which I believe require a minor revision to clarify

i) It is unclear whether the edge set is provided as prior knowledge, generated dynamically by the meta-LLM, or assumed to be fully connected. This ambiguity does not affect the current experiments, since there are only three nodes, one may assume a fully connected structure. But for larger numbers of agents, the construction of the edge set becomes an important design choice and should be explicitly defined.

ii) Clarification on Equation (4).  The response  $R_j'$ depends on all the nodes with higher influence score. Then should this concetenation be from $i=1$ to $i=j-1$? Also, what does it mean by $A_{ij} R_i^{Sorted}$? Is it a kind of normlization by $A_{ij}$? The notation $R_i^{Sorted}$ also looks strange, is it a output from node $i$?

**Reviewer Concerns:**

I believe most the concerns are addressed, while there are some points that still require clarification. However, all newly added experiments are still conducted on a small set of LLM agents. It remains unclear how the method would perform with a large pool of agents, and how the number $k$ should be chosen in such settings. These questions remain open.

**Reviewer Scores:**

Reviewer 4HpR indicated that he will raise his score to 6, and I agree with this adjustment.

Reviewer 47YL's score will remain 6.

For reviewers 7qjd and Usox, their concerns appear to be at least partially addressed. It is possible that each of them may increase their score by 1.

---

### Decision · Program_Chairs · 2026-01-26

Accept (Poster)